# Enantioselective OTUD7B fragment discovery through chemoproteomics screening and high-throughput optimisation
Aini Vuorinen [1,2,4], Cassandra R. Kennedy [2,4], Katherine A. McPhie [2,4], William McCarthy [2], Jonathan Pettinger [3], J. Mark Skehel [1], David House [3], Jacob T. Bush [3] ✉ & Katrin Rittinger [2] ✉

Deubiquitinating enzymes (DUBs) are key regulators of cellular homoeostasis, and their dysregulation is associated with several human diseases. The ovarian tumour protease (OTU) family of DUBs are biochemically well-characterised and of therapeutic interest, yet only a few tool compounds exist to study their cellular function and therapeutic potential. Here we present a chemoproteomics fragment screening platform for identifying novel DUB-specific hit matter, that combines activity-based protein profiling with high-throughput chemistry direct-to-biology optimisation to enable rapid elaboration of initial fragment hits against OTU DUBs. Applying these approaches, we identify an enantioselective covalent fragment for OTUD7B, and validate it using chemoproteomics and biochemical DUB activity assays.

Deubiquitinating enzymes (DUBs) are specialised proteases which remove ubiquitin post-translational modifications to regulate the ubiquitin system. Most cellular processes rely on reversible ubiquitination, so dysregulation of the ubiquitin machinery is often a driver for many different diseases. There are around 100 known human DUBs, split into seven different subfamilies: the distinct Zn-dependent metalloproteases (JAMMs); and the cysteine proteases ubiquitin C-terminal hydrolases (UCHs), ubiquitin-specific proteases (USPs), Machado-Joseph domain-containing proteases (MJDs), motif-interacting with ubiquitin containing proteases (MINDYs), zinc finger containing ubiquitin peptidase (ZUP), and the ovarian tumour proteases (OTUs)[1]. Novel, selective tool compounds are needed to better understand their cellular function and validate their therapeutic tractability.

While progress in the development of chemical probes for DUBs has been made in recent years, their translation into clinically-relevant inhibitors remains slow[2]. This may in part be due to challenges in designing selective DUB inhibitors, as many DUB catalytic pockets are structurally similar[3]. Nevertheless, a number of selective inhibitors have been reported for the USP subfamily, namely USP7[4–6], USP9X[7], USP25/28 dual inhibitors[8], and USP30[9]. The druggability of the UCH DUBs has also been validated with the discovery of a selective inhibitor for BAP1[10] and covalent cyanopyrrolidine tool compounds for UCHL1[11,12]. In fact, UCHL1 has been the target of several successful covalent chemical probe discovery campaigns,

where optimised cyanopyrrolidine and chloroacetamide activity-based probes have been used to characterise the mechanisms of deubiquitination and the downstream effects of UCHL1 inhibition in live cells and zebrafish embryos[13–15]. Furthermore, in the OTU subfamily, OTUB1 and OTUB2 have been liganded by covalent fragments using ABPP hit-finding strategies[16,17], and more recently, a potent and selective chloroacetamide inhibitor of VCPIP1 was reported[18]. A series of cyanopyrrolidine inhibitors of OTUD7B have also been disclosed in patents from Mission Therapeutics[19,20], and a non-covalent OTUD7B inhibitor has also been reported, however the compound structure has not been published[21].

OTUs constitute the second largest subfamily of DUBs, comprising 17 OTUs, differentiated by size of their catalytic OTU domain[22]. Of these, 16 have a catalytic cysteine. The otubains (OTUB1 and OTUB2) and the OTUDs (OTUD1, OTUD2, OTUD3, OTUD4, OTUD5, OTUD6A, OTUD6B, and ALG13) have smaller catalytic OTU domains, while the A20-like OTUs (OTUD7A, OTUD7B, A20, ZRANB1, and VCPIP1) and OTULIN have larger catalytic domains. The OTU family exhibit different ubiquitin chain cleavage selectivity, with OTUB1 cleaving K48-linked chains[22], OTULIN cleaving M1-linked poly-Ub chains[23], and OTUD7B (or Cezanne) specifically cleaving K11-linked polyubiquitin chains[24]. A number of the OTU DUBs, including A20 and OTULIN, are reported to negatively regulate immune responses through the NF-κB pathway[25]. Furthermore,

[1]Proteomics Science Technology Platform, The Francis Crick Institute, London, UK. [2]Molecular Structure of Cell Signalling Laboratory, The Francis Crick Institute, London, UK. [3]Crick-GSK Biomedical LinkLabs, GSK, Stevenage, Hertfordshire, UK. [4]These authors contributed equally: Aini Vuorinen, Cassandra R. Kennedy, Katherine A. McPhie. ✉e-mail: jacob.x.bush@gsk.com; Katrin.Rittinger@crick.ac.uk

several members of the OTU family are reported to be upregulated in cancer cells, including OTUD7B which deubiquitinates several substrates including oestrogen receptor α (ERα)[26], and GβL (a sub-unit of the mTOR complex)[27], thus promoting carcinogenesis. As such, inhibition of these OTU DUBs may be an interesting therapeutic strategy.

Fragment-based screening offers an attractive approach to identify starting points for OTU DUB tool compound development. Typically, fragment hits have higher ligand efficiencies than small molecule hits[28], meaning that good physicochemical properties can be maintained during fragment hit-to-lead optimisation. However, this still requires time- and resource-intensive medicinal chemistry campaigns[29]. Fragment screening also comes with inherent limitations as a result of their smaller size (<300 Da); fragment hit detection can be more challenging due to weaker affinities of fragments[28,30,31], and is often limited by compound solubility. This can be overcome by appending a reactive functional group to the fragment, which forms a covalent bond with nucleophilic residues on the protein surface, and increases the ease of hit detection[32]. We and others have made significant gains in chemoproteomics technologies for screening covalent fragments against endogenous cysteine-containing proteins to identify selective hits within cells and lysates[33–36]. When coupled with protein family-specific activity-based probes, such as DUB-specific probes, enriched samples are clean enough to be screened in a high-throughput manner[18,37]. We wanted to expand on our previous work[37], and enable elaboration of fragment hits by employing a high-throughput chemistry direct-to-biology platform to quickly progress initial fragment hits into more potent and selective inhibitors.

In this study, we translate these technological advances into inhibitor discovery, combining our optimised label-free chemoproteomics platform with high-throughput chemistry direct-to-biology optimisation (HTC-D2B), to screen, validate and optimise covalent fragment hits against DUBs. We present the unbiased screening of a library of 227 diverse chloroacetamide fragments in cell lysates, followed by targeted HTC-D2B validation and optimisation of fragment hits against five members of the OTU DUB family. We identify an enantioselective covalent fragment inhibitor of OTUD7B primed for medicinal chemistry optimisation efforts.

## Results and discussion

We employed our previously developed high-throughput chemoproteomics platform to screen a library of 227 cysteine-reactive chloroacetamide fragments (200 μM) against DUBs in HEK293T cell lysate[37]. Briefly, our platform employs competitive activity-based protein profiling (ABPP), where a DUB specific ABPP probe—biotinylated ubiquitin vinyl sulfone (Biotin-Ahx-Ub-VS)—is used to enrich for ubiquitin-binding proteins (Fig. 1A). By comparing fragment treated samples and DMSO controls, competition ratios for each fragment-DUB interaction are determined. In order to enable screening of larger fragment libraries, we increased the platform throughput by reducing the liquid-chromatography mass spectrometry (LC-MS/MS) instrument time from previously reported

**Fig. 1 | Chemoproteomics cysteine reactive fragment screening for deubiquitinating enzymes (DUBs). A** Schematic of competitive chemoproteomics platform workflow. Created in BioRender. Kennedy, C. (2024) https://BioRender.com/n38w715; **B** Summary of chemoproteomics fragment screening results, showing liganded DUBs by family, colour coded by number of fragment hits. Full-length DUB sequences were aligned with COBALT[53] and subsequently visualised with iTOL[54]. Fragment screening was performed with technical replicates ($n = 3$ for compound treated samples, $n = 44$ for DMSO samples).

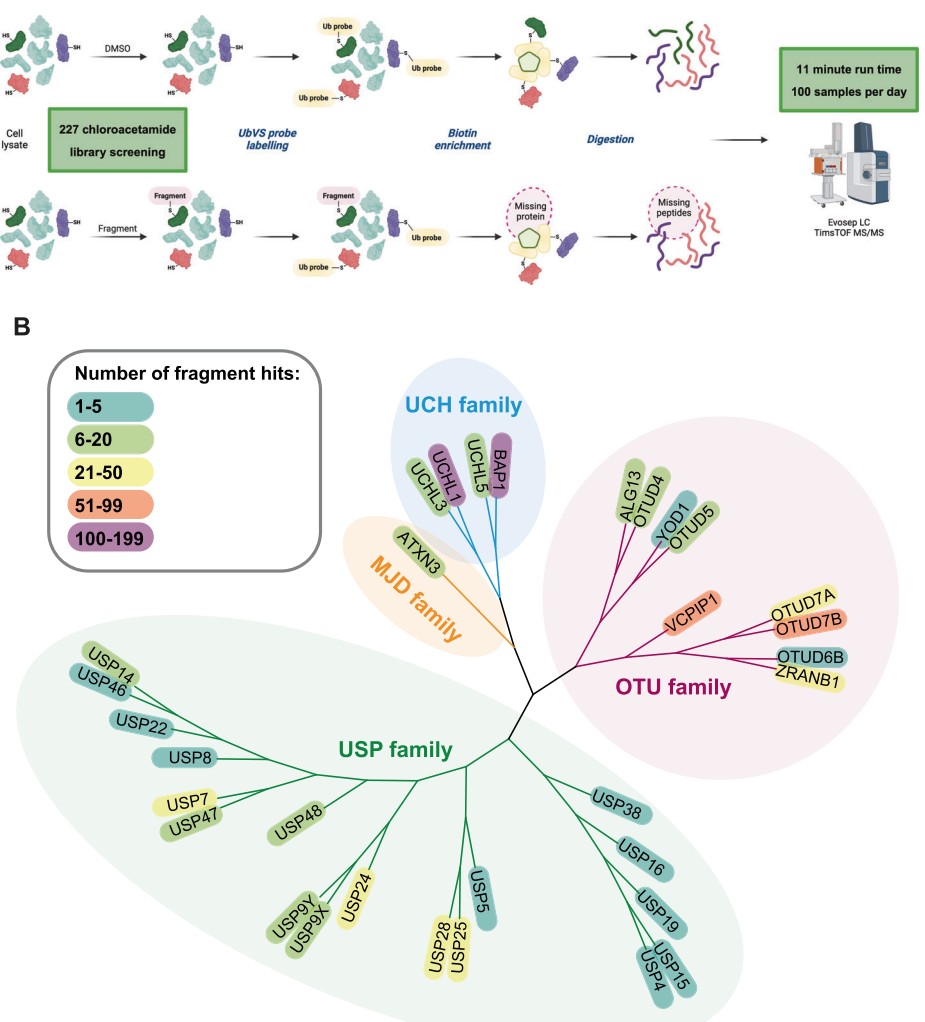

60 samples per day (SPD) with 21-min runs to 100 SPD (11-min runs). This was achieved by employing parallel accumulation-serial fragmentation combined with data-independent acquisition (diaPASEF) on an Evosep-timsTOF Pro 2 combination[38].

Our expanded fragment library consisted of a chemically diverse set of 227 chloroacetamide fragments (Supplementary Data 5) to cover a large chemical space (Supplementary Fig. 1A). A chloroacetamide electrophile was selected as our previous work demonstrated that acrylamide and methyl acrylate electrophiles exhibited poor activity toward DUBs[37]. A similar observation for an acrylamide electrophile was also recently reported by Chan et al.[18]. Although the more intrinsically reactive chloroacetamide electrophile[39] may not be optimal for clinical translation, there have been several examples in literature demonstrating that selectivity can be achieved with the chloroacetamide electrophile[36,40]. Moreover, our aim was to develop potent tool compounds using a fragment-based approach, and employing the chloroacetamide electrophile provides a useful avenue into hit identification where fragments have modest reversible interactions.

Using our higher throughput chemoproteomics platform, we quantified Biotin-Ahx-Ub-VS probe interactions with 43 DUBs covering MJD, OTU, UCH and USP subfamilies (Supplementary Fig. 2, Supplementary Data 1). No DUBs from the ZUP or MINDY subfamilies were identified. The JAMM family are excluded from our analyses as they do not have a catalytic cysteine. We hypothesise that the Biotin-Ahx-Ub-VS probe biases DUB identification away from families which preferentially recognise longer ubiquitin chain motifs and cleave polyubiquitin from the distal end, such as the MINDY family[41]. Our previous platform quantified interactions with 52 DUBs[37], however we decided the modest reduction in profiling depth was worth the gain in throughput (100 SPD, 11-min runtime *vs* 60 SPD, 21-min runtime) for our larger library screen.

Of the 43 DUBs, we found hit fragments for 32 DUBs across the four different subfamilies (MJD, OTU, UCH and USP) (Fig. 1B and Supplementary Fig. 1B, C). To identify hits, we analysed competition ratios by comparing the protein intensities of fragment treated samples to DMSO treated samples with an unpaired t-test. Hits were defined using the following cut offs: q-value ≤ 0.05, average $log_2$ ratio (fragment/DMSO) ≤ -1 and unique peptides ≥2. Overall, 95% of fragments competed at least one DUB (Supplementary Fig. 1B). This is consistent with labelling rates observed for chloroacetamide fragments in our previous screen (97%)[37]. The majority of fragments (63%) formed selective interactions (*i.e.* 1–3 DUB targets) whilst only a small number of fragments (5%) were highly promiscuous (*i.e.* >10 DUB targets). We observed no common pharmacophore represented in promiscuous fragments, nor enrichment of any features expected to enhance reactivity or lipophilicity. Inversely, we also observed that UCHL1 and BAP1 were both targeted by the majority of fragments (>140 fragment hits) (Supplementary Fig. 1C) perhaps due to the presence of a highly reactive catalytic cysteine. As both already have well-established tool compounds[10–12,42], we ruled them out for further follow-up.

Instead, we directed our efforts for follow-up work towards the OTU DUB family. Currently, many members of the OTU DUBs have yet to be targeted by well-characterised chemical tools, so we endeavoured to identify new tools that could interrogate OTU biology and therapeutic potential. We quantified the majority of OTU DUBs with a catalytic cysteine in our proteomics screen (11 of 16), identifying fragment hits against nine of them (Fig. 2A). We observed that some of these fragment hits labelled one OTU DUB selectively, while other fragments labelled several OTU DUBs with minimal labelling of other DUB families. Crucially, none of the fragment hits against OTU DUBs labelled more than 15 other non-OTU DUB family proteins, which suggested good protein ligandability without extensive fragment promiscuity (Supplementary Fig. 3A–C).

We selected fragment hits for in vitro validation and optimisation based on three main selection criteria:

1. Average $log_2$ ratio (fragment/DMSO) < -1.5 (or < -1.1 if other criteria met);
2. Fewer than five off-target DUB proteins;

3. Chemical structure amenable to high-throughput chemistry (aromatic amines were discounted due to low conversion rates in amide couplings).

Seven fragments were selected for further investigation (Fig. 2B and Supplementary Fig. 4A, B) against five OTU family proteins: OTUD4, OTUD5, OTUD7A, OTUD7B and ZRANB1. The selected fragments included compound **2**, which was profiled as an OTUD7B inhibitor in our previous work (therein compound **26**)[37]. A dilution series of compounds **1**–**7** was incubated for 24 h with recombinantly expressed OTUs, and labelling was measured by intact protein LC-MS (Supplementary Fig. 5A, B and Supplementary Fig. 6). In vitro labelling of OTUD4, OTUD5, OTUD7B, and ZRANB1 by all seven fragments reflected labelling observed in our chemoproteomics screen, with the exception of one fragment which labelled ZRANB1 by intact protein LC-MS but not by chemoproteomics. For OTUD7A, chemoproteomics and intact protein LC-MS were less concordant, with some discrepancies between the two techniques observed for several fragments labelling OTUD7A (Fig. 2C). We observed that different DUBs reached differential maximum labelling percentages (Supplementary Fig. 5A, B), suggesting heterogeneity between protein batches, and therefore direct comparisons between labelling percentages across DUBs were treated with caution. However, the differentiated binding profiles between specific fragments for each given DUB is striking and demonstrates that selective interactions are forming. The notable difference in labelling for OTUD7A in vitro and in lysate proteomics may suggest that its catalytic cysteine is regulated in a cellular context, perhaps by another protein, co-factor or post-translational modification, that has yet to be reported in literature. OTUD5 is known to be activated in cells through phosphorylation, altering its propensity to bind ubiquitin[43]. To investigate whether we had selective fragment binders for either form of OTUD5, we validated both the inactive and activated forms, however we observed similar labelling for both states (Supplementary Fig. 5C), supporting that OTUD5 phosphorylation does not alter the nucleophilicity of the catalytic cysteine.

To rapidly elaborate compounds **1**–**7** toward more potent and selective binders we used a high-throughput chemistry direct-to-biology screening platform (HTC-D2B) to generate and screen related fragments (Fig. 3A). The optimisation of fragments in hit-to-lead medicinal chemistry campaigns is still a bottleneck in developing potent and selective tool compounds[29]. In our HTC-D2B platform[44,45], we used a single-step amide coupling reaction to install the chloroacetamide electrophile in a 384-well plate format. Upon quenching the reaction, we screened crude reaction mixtures without purification against recombinant protein, enabling rapid generation of structure-activity relationship data. Although product purities are not uniform across the plate, the HTC-D2B platform facilitates informed decision-making to select elaborated fragments for resynthesis, purification and validation.

We designed a library of 351 amines around the structures of fragment hits **1**–**7** using Tanimoto similarity constraints[46,47]. Amine selection was restricted to compounds that had a molecular weight between 110 and 350 Da and were readily available. HTC was performed in situ in a 384-well plate, creating a secondary library of 351 cysteine-reactive chloroacetamide fragments. Following analysis by LC-MS to determine conversion rates[48] (Supplementary Fig. 7A, B), and a reagent quench with hydroxylamine (where tolerated, see below), the secondary HTC library was directly incubated with the five recombinant OTU DUBs of interest and protein labelling measured as before by intact protein LC-MS (Supplementary Fig. 7C).

OTUD4, OTUD5 and OTUD7A tolerated exposure to hydroxylamine well, however we observed a loss of MS signal for OTUD7B and ZRANB1, and screening of the HTC library was therefore repeated against these two proteins without the reagent quench step. When compared to round 1 validated fragments, we observed a notable improvement in labelling across the library for OTUD7B and ZRANB1, several improved compounds for OTUD4 and OTUD7A, but no significant improvement for OTUD5 (Fig. 3B). We selected the most improved compounds from HTC-D2B

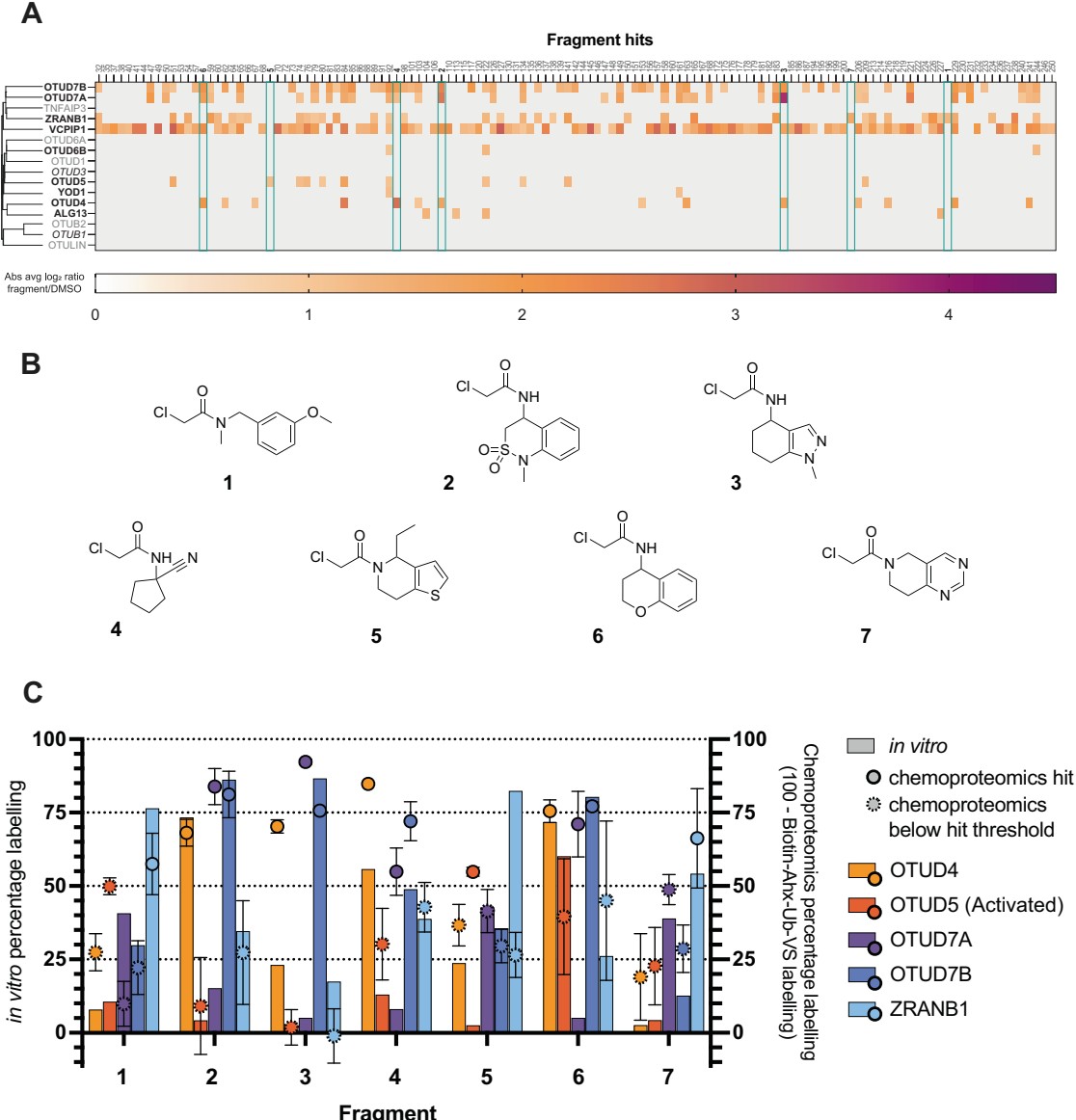

**Fig. 2 | Fragment hit identification and validation. A** Fragment hits (screened at 200 μM, $n = 3$ for compound treated samples, $n = 44$ for DMSO samples) against OTU family members showing competed DUBs (bold), quantified DUBs (italics) and DUBs not identified (grey). Average $\log_2$ ratios are plotted as absolute values for simplicity. Full-length DUB sequences containing catalytic site cysteines were aligned using COBALT[53]. Fragment hits against other DUB subfamilies are presented in Supplementary Fig. 3; **B** Structures of selected hits against OTU family DUBs: compounds **1**–**7**; **C** Validation of compounds **1**–**7** (200 μM) comparing

in vitro validation of recombinant OTU proteins (% labelling, bar chart) with chemoproteomics (100 - Biotin-Ahx-Ub-VS % labelling (fragment/DMSO), $n = 3$ for fragment treated samples, $n = 44$ for DMSO controls, dot plot; dotted points represent values below the hit threshold; error bars represent SEM). OTUD4 (orange), activated OTUD5 (red), OTUD7A (purple), OTUD7B (dark blue), ZRANB1 (light blue). Representative examples of deconvoluted intact protein LC-MS spectra for compound **6** are presented in Supplementary Fig. 6.

experiments against each OTU DUB for further validation and proteomics selectivity studies. To avoid comparisons between labelling events with and without the hydroxylamine quench step, we focused on compounds that showed improved labelling for a given protein at this stage, rather than comparing fragment labelling across all five proteins. We considered HTC compounds to be a hit if they showed higher protein labelling efficiency than the original fragment hits, and we excluded any hits that showed multiple labelling events on a single protein. Based on these criteria, we selected 21 compounds for retesting (**8**–**28**, Supplementary Fig. 8 and Supplementary Data 5). Compounds **11**–**28** were purchased as purified compounds, and taken forward alongside the original hits **1**–**7** for proteomics profiling in

lysates to determine on-target engagement and selectivity across the DUBome.

For round 2 chemoproteomics profiling, HEK293T cell lysates were treated with 25 chloroacetamides (compounds **1**–**7**, **11**–**28**; 7 original fragment hits, and 18 purified hits from HTC-D2B screen) at 50 μM and 200 μM (Supplementary Data 2). We observed a similar number of DUBs enriched when compared to the round 1 screen (Supplementary Fig. 9A), and good reproducibility of the seven original fragment hits (Supplementary Fig. 9B), however OTUD5 was not detected in this chemoproteomics experiment (Supplementary Fig. 9A). We were pleased to observe HTC-D2B optimised compounds that were at least as potent as our initial hits, and

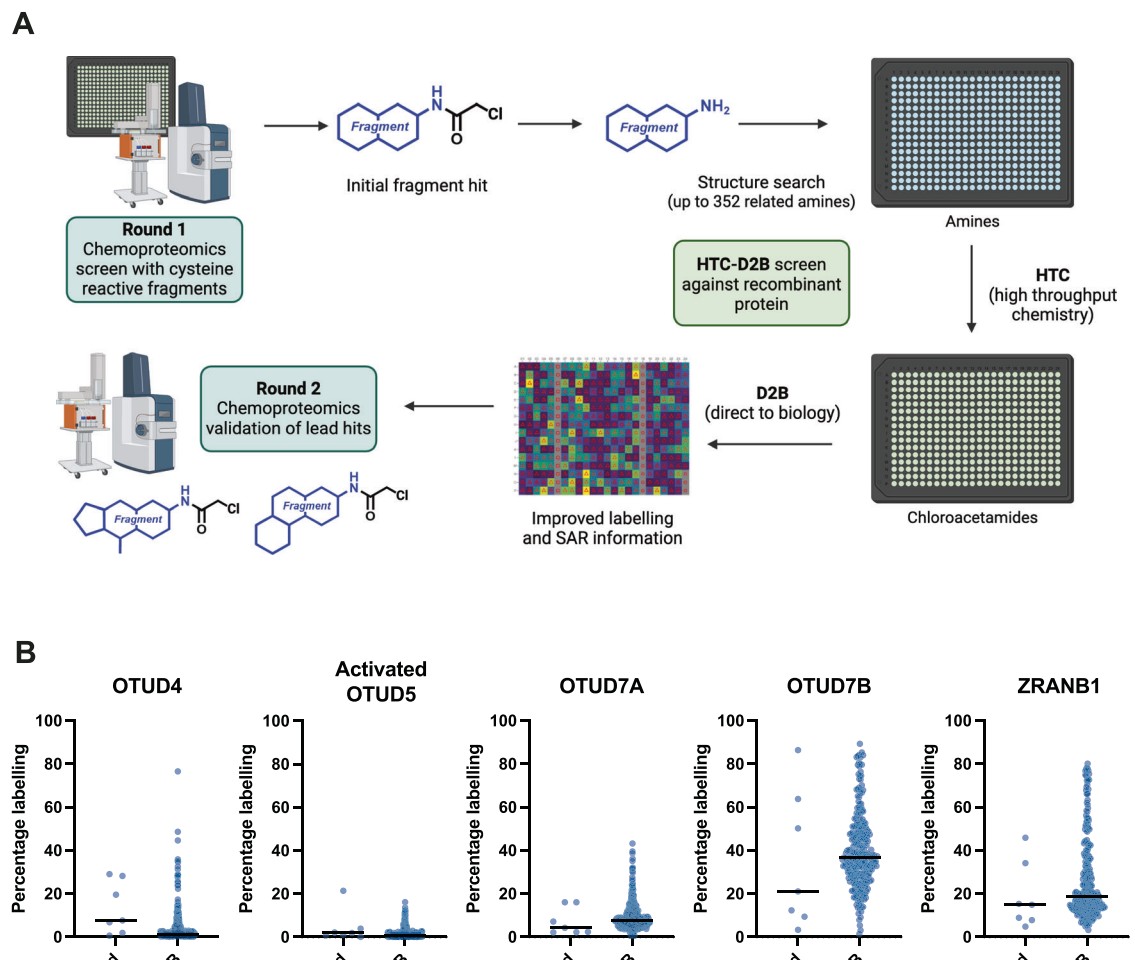

**Fig. 3 | High-Throughput Chemistry Direct-to-Biology (HTC-D2B) screening.**
**A** Schematic of the workflow for HTC-D2B screening[45]. The initial fragment library was screened by chemoproteomics (round 1), with hits identified from this screen used as inputs to design an HTC library of 351 structurally-related amines. HTC was performed to install the chloroacetamide electrophile, and the library screened D2B with no purification against recombinant protein. Created in BioRender. Kennedy, C. (2024) https://BioRender.com/s90p338; **B** Percentage labelling of recombinant proteins by HTC-D2B library at 50 μM compared to percentage labelling of purified round 1 hits at 50 μM. Median labelling represented by black line.

beyond the upper limit of detection at the concentrations employed in round 2 profiling for our OTU proteins of interest OTUD7B (Fig. 4A), OTUD4, OTUD7A and ZRANB1 (Supplementary Fig. 9C).

We were intrigued by compound **28**, which showed improved labelling of OTUD7B over the initial fragments **2** and **6** (from which it was derived), and was at least as potent as initial fragment **3** (Fig. 4A). We further noted that **28** showed impressive selectivity towards OTUD7B over all other DUBs (Fig. 4B), and that it was a racemic compound. We were interested to see if the two enantiomers of **28** showed different labelling profiles, so we decided to interrogate the potency window between the two enantiomers (**29** and **30**) (Fig. 4C). A concentration-response chemoproteomics experiment confirmed enantioselective targeting of OTUD7B with the active *S*-enantiomer (**29**, TE50 (concentration required for 50% target engagement) = 3.8 μM), which was ~13-fold more active than the less active *R*-enantiomer (**30**, TE50 = 50.5 μM) (Fig. 4D, Supplementary Fig. 10A and Supplementary Data 3). Furthermore, we demonstrated ~8-fold selectivity of OTUD7B over OTUD4 for compound **29**, and even greater selectivity over BAP1, VCPIP1 and USP28 (Fig. 4E and Supplementary Fig. 10A–C). Due to identification of a small number of low intensity peptides for OTUD7A, we were unable to measure accurate TE50 values for OTUD7A by chemoproteomics, and therefore could not compare compound selectivity between OTUD7B and OTUD7A by

chemoproteomics. Other DUBs showed no concentration-dependent fragment engagement.

In order to identify non-DUB off-targets of enantiomers **29** and **30**, we investigated their proteome-wide cysteine reactivity using an iodoacetamide-desthiobiotin (IA-DTB) competitive chemoproteomics workflow[35]. HEK293T cell lysates were treated with fragments **29** and **30** at 50 μM and 200 μM, followed by treatment with IA-DTB probe. After trypsin digestion, the IA-DTB labelled peptides were enriched and analysed by LC-MS/MS. By comparing compound treated samples and DMSO controls, we determined competition ratios (CR; DMSO/compound) for IA-DTB labelled cysteine-containing peptides. At 200 μM, the enantiomers **29** and **30** labelled only 11 and 22 cysteine residues, respectively, across the proteome, and a concentration-dependent reduced response at 50 μM was observed for five and eight of those residues, respectively (Supplementary Fig. 10D). The catalytic site cysteine of OTUD7B (Cys194) was not detected by the IA-DTB probe (Supplementary Data 4). However, both enantiomers **29** and **30** showed similar labelling of OTUD4 catalytic site cysteine (Cys45), which was consistent with our concentration-response data using the Biotin-Ahx-Ub-VS probe.

We next validated the enantioselective activity of these compounds **28**, **29** and **30** with recombinant OTUD7A and OTUD7B using a fluorescence-based biochemical substrate assay[37] (Fig. 5A, Supplementary Fig. 11A and

**Fig. 4 | Proteomics validation of enantioselective OTUD7B fragment. A** Chemoproteomics validation of all fragments ($n = 3$ for fragment treated samples, $n = 26$ for DMSO controls) at 200 µM (black) and 50 µM (grey) against OTUD7B with compound **28** highlighted (hits against other OTU DUBs can be seen in Supplementary Fig. 9C); **B** DUBs labelled by compound **28** at 200 µM (black) and 50 µM (grey). DUBs that were not labelled by compound **28** are not shown ($n = 3$ for fragment treated samples, $n = 26$ for DMSO controls); **C** Structures of compounds **28** (racemate), **29** (S-enantiomer, active) and **30** (R-enantiomer, less active); **D** Full concentration-response of compounds **28** (grey), **29** (purple) and **30** (teal) against OTUD7B by chemoproteomics; **E** Concentration-response curves of compound **29** with OTUD7B, OTUD4, BAP1, VCPIP1 and USP28. Concentration-response data are presented as mean ± SEM, $n = 3$ for compound treated samples, $n = 15$ for DMSO controls. Some error bars are too small to be displayed. The curves were fitted with GraphPad Prism 10 using four parameter nonlinear regression with constraint bottom = 0, top = 100. 95% CI values are reported in Supplementary Fig. 10A.

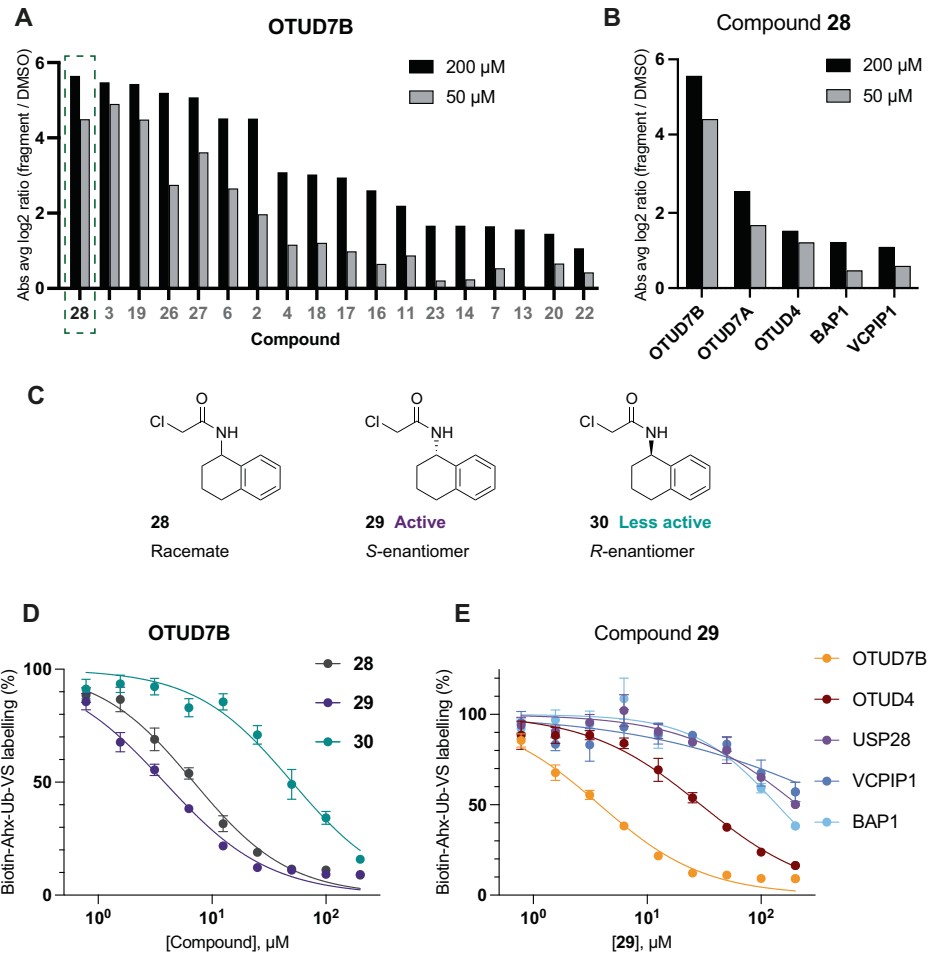

Supplementary Data 7). We observed the active S-enantiomer (**29**, IC50 = 3.5 µM) to be ~17-fold more inhibitory of OTUD7B activity than the less active R-enantiomer (**30**, IC50 = 58.3 µM). Interestingly, this observation concurs with enantioselectivity observed for a pair of published cyano-pyrrolidine OTUD7B inhibitors[19]. Furthermore, we saw selectivity of compounds **28**, **29** and **30** for OTUD7B over OTUD7A (~45-fold for compound **29**) in biochemical assays.

It is particularly notable that such a small fragment with minimal molecular complexity exhibits enantioselectivity, suggesting that fragment binding is not solely driven by chloroacetamide reactivity or fragment physical properties, but depends upon intrinsic complementarity of OTUD7B's catalytic site surface features and the 3D conformation of the fragment. Enantioselectivity is a common feature among both chemical tools and marketed drugs, and increasing consideration is being given to designing libraries of fragments with greater 3D-character and chiral diversity[49]. Furthermore, for compounds which display enantioselectivity, the less active enantiomer can be used as a logical negative control during hit-to-lead optimisation and biological validation. We observed that both enantiomers **29** and **30** had similar off-target profiles, validating their usefulness as opposing chemical tools (Supplementary Fig. 10B, C).

To further characterise the kinetics of irreversible inhibition of OTUD7B and OTUD7A by enantiomers **29** and **30**, we obtained the second-order rate constant of inhibition ($k_{inact}/K_I$). A dilution series of compounds **29** and **30** was incubated with recombinantly expressed OTUD7B and OTUD7A, and labelling was measured at 8 time-points over the course of 18 h by intact protein LC-MS (Fig. 5B, Supplementary Fig. 11B and Supplementary Data 8). The $k_{inact}/K_I$ values derived from these measurements strongly reinforced the enantioselectivity observed in

biochemical assay and by chemoproteomics. For OTUD7B inhibition by compound **29**, $k_{inact}/K_I = 5.5 \pm 0.1$ M⁻¹ s⁻¹, which was ~37-fold greater than for compound **30** ($k_{inact}/K_I = 0.15 \pm 0.004$ M⁻¹ s⁻¹) (Fig. 5C). Although, a plateau of covalent labelling for compound **30** was not reached after 18 h, the stark contrast between compounds **29** and **30** gives strong evidence of enantioselectivity. For OTUD7A, we observed minimal labelling for both compounds **29** and **30** across 18 h, even at the highest concentration tested (100 µM) (Supplementary Fig. 11B). As labelling of OTUD7A was so low, and in the linear phase, we were unable to derive $k_{inact}/K_I$ values for OTUD7A. Again, this observation reinforced the selective inhibition of OTUD7B over OTUD7A by the enantiomers that we observed biochemically.

Next, to confirm inhibition of OTUD7B activity was caused by compound binding at the catalytic cysteine, we performed site identification (SiteID) experiments to identify the site of covalent labelling by compound **29**. Recombinant OTU domain of OTUD7B was incubated with compound **29**, digested with trypsin and resulting peptides were analysed by LC-MS/MS. We observed that OTUD7B was selectively labelled with a single modification at the catalytic cysteine residue, Cys194 (Fig. 5D).

Finally, we turned to structural biology to understand why S-enantiomer **29** is more active than R-enantiomer **30**. Attempts to crystallise OTUD7B with these fragments were unsuccessful, so we used covalent molecular docking of compound **29** (active enantiomer, purple) and compound **30** (less active enantiomer, teal) into OTUD7B OTU domain (PDB 5LRU[24]) at catalytic residue Cys194 to provide further insight into differences in binding and activity observed between the two enantiomers. Docking predicted a 180° conformational flip between the active enantiomer **29** (Supplementary Fig. 12A) and the less active enantiomer **30**

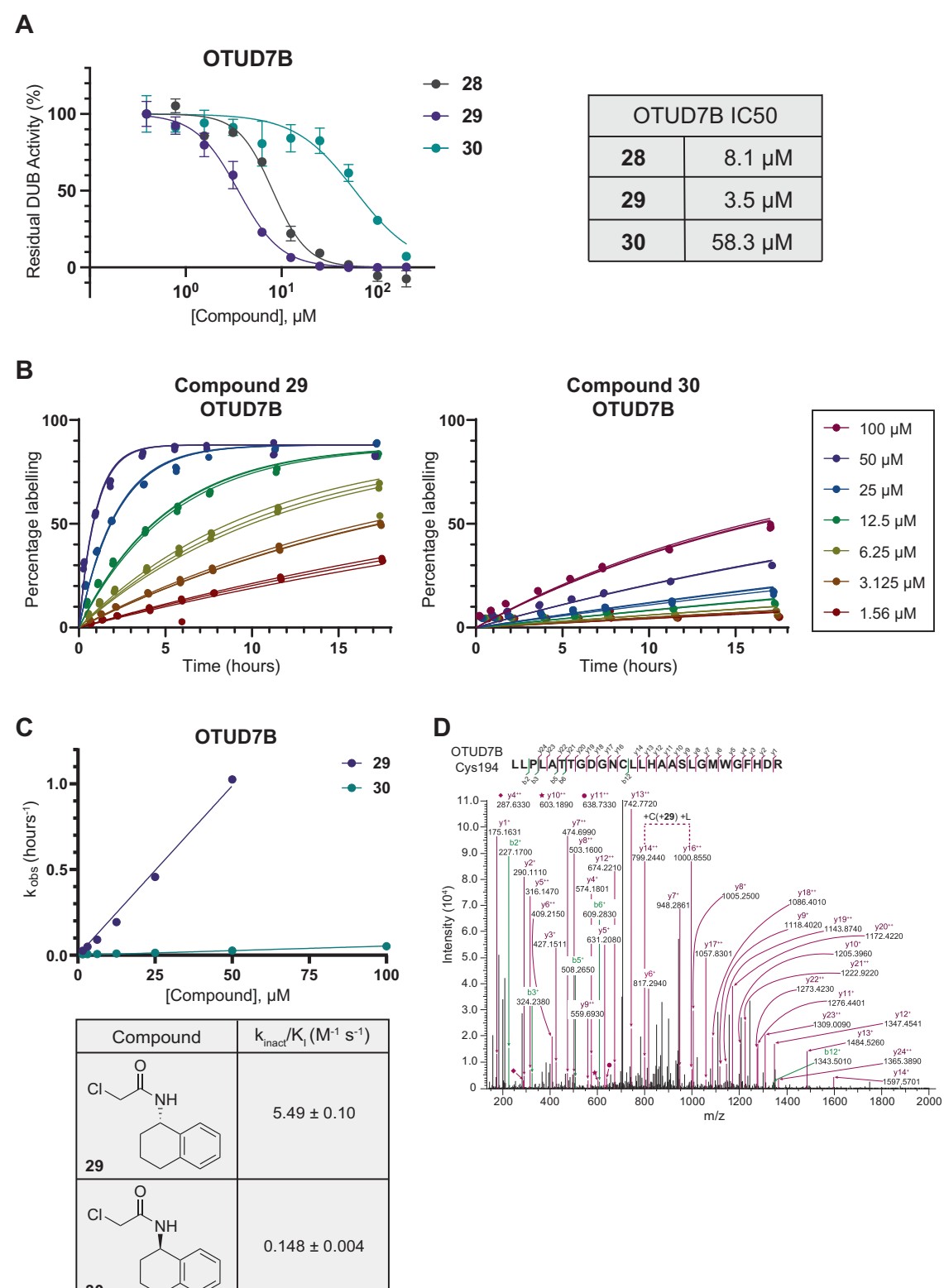

(Supplementary Fig. 12B), which alters the placement of the electrophilic amide and aromatic ring within the pocket. However, visualisation of these two predicted conformations (Supplementary Fig. 12C, D) and a 2D interaction map (Supplementary Fig. 12E, F) did not highlight any major differences in the types of interactions formed within the binding site. As such, although this covalent molecular docking suggests a possible binding mode, structural biology would be required to further elucidate compound

mechanism of action and explain the differences in potency observed between the two enantiomers.

At present, the enantiomers are relatively small fragments with MW < 300, and electrostatic interactions between the enantiomers and side chains of the protein are limited. However, the enantioselectivity discovered in this study provides an excellent starting point for further optimisation to develop more potent and selective tool compounds for OTUD7B.

**Fig. 5 | Lead compound validation. A** Enzymatic inhibition assay data for compounds **28** (grey), **29** (purple) and **30** (teal) against OTUD7B. Data are presented as mean ± SD, $n = 3$. Some error bars are too small to be displayed. The curves were fitted with GraphPad Prism 9 using four parameter nonlinear regression with constraint bottom = 0, top = 100. IC50 values are quoted to 1 decimal place, and 95% CI values are reported in Supplementary Fig. 11A; **B** Time courses (0–18 h) of compound labelling (100–1.56 μM) against OTUD7B (0.5 μM), performed in technical triplicate (shown on graphs). Labelling percentages were plotted against time in GraphPad Prism 10, and curves were fitted separately for each replicate using one-phase association, with constraints Y0 = 0 and plateau = highest labelling percentage; **C** Pseudo-first order rate constant values ($k_{obs}$) from time course labelling graphs were plotted against concentration, in triplicate, and fitted using straight line fit, with constraints $Y_{intercept} = 0$. Data are presented as mean ± SD, $n = 3$. Error bars are too small to be displayed. Slope of straight line fit was converted from $\mu M^{-1}\ h^{-1}$ to $M^{-1}\ s^{-1}$ to give reported $k_{inact}/K_I$ values. For compound **29**, 100 μM $k_{obs}$ value was not used to calculate $k_{inact}/K_I$, as it was outside the linear range. Errors are reported as standard error, as calculated in GraphPad Prism 10; **D** LC-MS/MS spectra of OTUD7B peptide $_{183}$LLPLATTGDGNCLLHAASLGMWGFHDR$_{209}$ labelled by compound **29** indicating Cys194 as the site of compound labelling.

## Conclusions

We have increased the throughput (100 SPD) of our previously established high-throughput chemoproteomics platform (60 SPD) for the identification of novel covalent molecules targeting the DUB family of ubiquitin system enzymes. Using this platform, we screened a library of 227 chloroacetamide fragments and identified hits against 32 DUBs across the MJD, OTU, UCH, and USP subfamilies. We selected covalent fragment hits against the OTU subfamily for further validation, as this subfamily has therapeutic relevance and a limited number of tool compounds currently available. We applied high-throughput chemistry direct-to-biology (HTC-D2B) screening to rapidly test elaborated analogues of covalent fragment hits against recombinant protein. The risk of purified protein screening is that improved labelling can be driven by increased fragment reactivity and/or lipophilicity, which does not correlate with increased potency or selectivity in a cellular context. However, we were pleased to see that employing HTC-D2B screening and chemoproteomics in partnership in this study mitigated this risk. Exciting future developments enabling HTC-D2B screening directly in cell lysates will go even further, allowing faster and more efficient access to compound SAR and selectivity data in a cellular context.

Following our HTC-D2B screening, we observed improved labelling of the OTU DUB, OTUD7B with racemic compound **28** over its parent hits **2** and **6**, and further interrogated the activity of its two enantiomers. We identified an enantioselective inhibitor of OTUD7B, compound **29** (TE50 = 3.8 μM, ~13-fold more active than opposite enantiomer), representing a novel pharmacophore and the first use of a chloroacetamide electrophile for OTUD7B inhibition. We confirmed the selectivity window of our inhibitor, compound **29**, by proteomics, and demonstrated selectivity over other OTU subfamily members (~8-fold), as well as other DUBs (>36-fold). Biochemical activity assays and kinetic characterisation confirmed the enantioselectivity of compound **29** (IC50 = 3.5 μM and $k_{inact}/K_I = 5.5 \pm 0.1\ M^{-1}\ s^{-1}$, ~17-fold more active and ~37-fold greater than opposite enantiomer, respectively), and demonstrated selectivity over a structurally related DUB, OTUD7A (~45-fold selectivity for OTUD7B over OTUD7A). Finally, we confirmed that compound **29** binds at the catalytic residue Cys194 of OTUD7B, and we modelled the potential binding modes of the two enantiomers using covalent docking into the crystal structure of OTUD7B's OTU domain. Although compound **29** has low molecular complexity and modest affinity for OTUD7B, we observed selectivity for OTUD7B labelling under biologically relevant conditions.

We therefore propose that compound **29** would be a good starting point for medicinal chemistry to create a potent and selective inhibitor for OTUD7B, particularly supported by the availability of X-ray crystal structures of its OTU domain[24]. With OTUD7B substrates including ERα[26], GβL[27], EGFR and Sox2[50,51], identifying potent OTUD7B inhibitors would be very valuable for therapeutic translation.

## Methods

Fragment library compounds can be found in Supplementary Information (Supplementary Data 5).

## Cell lines

HEK293T cell line (female human origin, ATCC, Cat# CRL-3216) was used in this study. The cells were maintained at 37 °C with 5% $CO_2$ in DMEM media supplemented with 10% fetal bovine serum and 1% L-Glutamine–Penicillin–Streptomycin solution (200 mM L-glutamine, 10,000 U/mL penicillin and 10 mg/mL streptomycin). Prior to lysis, cells were pelleted, washed twice with PBS and frozen. Cells were authenticated by STR profiling by Francis Crick Institute Cell Services STP. All cells were negative from mycoplasma contamination tested by fluorescence staining, agar culture, and PCR testing by Francis Crick Institute Cell Services STP.

## DUB chemoproteomics experiments

Chemoproteomics samples were prepared as previously described[37].

Briefly, HEK293T cells were lysed in lysis buffer containing 50 mM TRIS pH 7.5, 150 mM NaCl, 0.5% CHAPS hydrate, 0.1% IGEPAL CA-360, 5 mM $MgCl_2$, 10% Glycerol and protease inhibitor cocktail (Sigma-Aldrich; P8340). Lysates (495 μg per sample) were treated in triplicate with fragments at 200 μM or DMSO for 3 h followed by 1 h treatment with Biotin-Ahx-Ub-VS (UbiQ; UbiQ-188) at 0.5 μM final concentration. Labelled proteins were enriched using acetylated NeutrAvidin slurry (Thermo Scientific; 29204). Proteins were reduced and alkylated and then digested on-bead with LysC (Wako; 125-05061). The supernatants were collected and further digested with trypsin (Thermo Scientific; 90057) and acidified with formic acid.

The samples and iRT standard (Biognosys; Ki-3002-2) were loaded onto Evotips (Evosep; EV2011), which were prepared according to manufacturer's instructions. The peptides were analysed using an Evosep One LC system (Evosep) coupled with a timsTOF Pro 2 (Bruker) mass spectrometer via a CaptiveSpray nano-electrospray ion source. Data for all samples was acquired in diaPASEF mode using the 100 SPD predefined method on Evosep One, which was fitted with an 8 cm column (Evosep; EV1109). Mobile phase A was 0.1% formic acid in water and mobile phase B 0.1% formic acid in acetonitrile.

For all samples, mass spectra were acquired from 100 to 1700 $m/z$. The ion mobility range was set to 0.6–1.68 Vs $cm^{-2}$. TIMS accumulation and ramp times were set to 100 ms. Eight diaPASEF scans were collected per one TIMS-MS scan, giving a duty cycle of 0.95 s. 24 fixed mass windows of 25 $m/z$ were set over the mass range 400–1000 $m/z$ and mobility range 0.6–1.64 Vs $cm^{-2}$. The collision energy was increased linearly from 20 eV to 59 eV between 0.6 and 1.60 Vs $cm^{-2}$.

The data were searched using Pulsar search engine in Spectronaut (v.16). A spectral library was first generated by searching the DIA data against human Uniprot, contaminants and avidin fasta files. BGS factory settings (default) were used. For the Round 2 chemoproteomics experiments combined libraries with Round 1 library were created.

The data was searched against the generated library. QUANT 2.0 was selected as Protein LFQ Method. The data was normalised using global median normalisation strategy with automatic row selection. Run wise imputation with Q-value percentile = 10% was selected. Other search settings were used as default (BGS factory settings). Un-paired t-test was used to determine average $\log_2$ ratios (fragment/DMSO) and q-values. Before plotting all data was filtered for unique peptides ≥2. Significantly competed DUBs were defined based on average $\log_2$ ratio ≤ -1 and q-value ≤ 0.05.

Concentration-response curves were fitted using label free protein quantity values. Values were normalised against samples treated with DMSO and plotted as percentages of Biotin-Ahx-Ub-VS labelling against fragment concentration using GraphPad Prism (v. 10). The curves were fitted using four parameter nonlinear regression with constraints bottom = 0, top = 100. Each experiment was set up as technical triplicates.

### IA-DTB chemoproteomics experiment

IA-DTB proteomics was performed as previously described[35].

Briefly, HEK293T cells were lysed in RIPA lysis buffer (50 mM HEPES, pH 7.5, 150 mM NaCl, 1.0% IGEPAL CA-630, 0.5% sodium deoxycholate, 0.1% SDS) containing protease inhibitor cocktail (Sigma-Aldrich; P8340). The lysate was sonicated and clarified by centrifugation followed by filtration through a 0.22 µM filter. Protein concentration was determined using a BCA assay (Thermo Scientific, 23227) and diluted to 2.53 mg/mL with lysis buffer.

Lysates (500 µg per sample) were treated with fragments at 50 and 200 µM ($n = 4$) or DMSO ($n = 64$) for 4 h followed by 1 h treatment with iodoacetamide-desthiobiotin (IA-DTB) (Enamine; Z4994458618) at 500 µM final concentration. The lysates were then reduced and alkylated. The proteins were precipitated and digested with trypsin (Thermo Scientific; 90059) using a glass bead assisted sample clean up and digestion procedure.

Digested peptides were collected and enriched using NeutrAvidin beads (Thermo Scientific; 29204). The enriched peptides were eluted from the beads with 0.1% formic acid in 50% acetonitrile and dried overnight in Labconco CentriVap Benchtop Vacuum Concentrator at 4 °C. Samples were redissolved in 0.1% trifluoroacetic acid in water and desalted using C18 Nest desalting plates (The Nest Group Inc; HNS S18V) according to manufacturer guidance. Peptides were eluted and collected with 0.1% trifluoroacetic in 50% acetonitrile by centrifugation. The eluted peptides were dried overnight in Labconco CentriVap Benchtop Vacuum Concentrator at 4 °C.

Peptides were redissolved in 100 µL 0.1% formic acid in water, before loading the samples and iRT standard (Biognosys; Ki-3002-2) onto Evotips (Evosep; EV2011), which were prepared according to manufacturer's instructions. The peptides were analysed using an Evosep One LC system (Evosep) coupled with a timsTOF Pro 2 mass spectrometer (Bruker) via a CaptiveSpray nano-electrospray ion source. Data for all samples was acquired in diaPASEF mode using the 60 SPD predefined method on Evosep One, which was fitted with an 8 cm column (EV1109). Mobile phase A was 0.1% formic acid in water and mobile phase B 0.1% formic acid in acetonitrile.

For all samples, mass spectra were acquired from 100 to 1700 m/z. The ion mobility range was set to 0.6–1.60 Vs cm$^{-2}$. TIMS accumulation and ramp times were set to 100 ms. 12 diaPASEF scans were collected per one TIMS-MS scan, giving a duty cycle of 1.37 s. 24 variable mass and mobility windows were set over the mass range 400–1399.8 m/z and mobility range 0.6–1.60 Vs cm$^{-2}$ (Supplementary Data 6). The collision energy was increased linearly from 20 eV to 59 eV between 0.6 and 1.60 Vs cm$^{-2}$.

The data was searched using Pulsar search engine in Spectronaut (v. 19) against previously generated spectral library[35], where IA-DTB ($C_{14}H_{24}O_3N_4$, 296.18 Da) and carbamidomethyl were selected as variable modifications for cysteine residue. PTM workflow and localisation filter were selected. Other search settings were used as default (BGS factory settings). The precursor intensity data was exported. In order to obtain a single peptide-level intensity data and calculate competition ratios (CR = Intensity$_{DMSO}$/Intensity$_{compound}$), the subsequent analysis was performed using Python. Binding events were defined based on the following criteria: mean $\log_2(CR) \geq 1$ and $p$-value $\leq 0.05$. In addition, the peptide must have been detected in at least two of the compound-treated replicates and in ≥90% of all samples in the experiment. The peptide must have a coefficient of variation (CV) of ≤40% in the control samples. The peptide must only have a single IA-DTB modification and if multiple cleavage forms of the same peptide exist in the dataset, then only the most abundant of these peptides is considered.

**Table 1 | Deconvolution conditions for recombinant proteins**

| Protein construct | Expected mass range | *m/z* range |
|---|---|---|
| OTUD4 aa1-156 | 16000–20000 | 350–2000 |
| OTUD5 aa172-351 | 19000–23000 | 350–2000 |
| OTUD7A aa1-462 | 50000–55000 | 350–2000 |
| OTUD7B aa129-438 | 33000–38000 | 350–2000 |
| ZRANB1 aa343-692 | 39000–43000 | 350–2000 |

### Intact protein LC-MS

Recombinant OTU proteins were expressed and purified from BL21 *E.coli* cells (Agilent Technologies; cat #230132). The proteins were purified via GST or His-tag enrichment, before tag-cleavage with 3 C protease. The DUBs were further purified through ion exchange chromatography and size exclusion chromatography, before storage in 50 mM HEPES pH 7.5, 150 mM NaCl, 0.5 mM TCEP and 5% glycerol. OTUD5 was activated by treatment with CK2 kinase (New England BioLabs; P6010S) at 30 °C, 600 rpm for 4 h, and the reaction was quenched with 20 mM EDTA, as previously described[43].

1 µM OTUD4, OTUD5 or 0.5 µM OTUD7A, OTUD7B, ZRANB1 were incubated with 200-25 µM fragments for 24 h at 4 °C, in 25 mM HEPES pH 7.5, 50 mM NaCl buffer. Intact protein LC-MS was performed as previously described[37]. The deconvolution conditions for recombinant proteins studied in this work are presented in the Table 1.

### HTC-D2B

HTC-D2B was performed as previously described[45].

The HTC library of 351 parent amines was designed by using the parent amine SMILES strings of compounds **1** – **7** as inputs for structural similarity search. Structurally similar amines were searched within GSK solution and solid stocks, using criteria 110 < MW < 350, primary and/or secondary aromatic amines excluded, and phenols and tricyclic compounds excluded. Anilinic amines, tricyclic motifs and phenol-containing compounds are incompatible with the HTC reaction.

351 resulting amines were plated as 10 mM stock solutions in DMSO (20 µL, 1 eq.) in a 384-well plate. To each well containing amine, a solution of *N*-(Chloroacetoxy)succinimide (2 eq.) and *N,N*-Diisopropylethylamine (3 eq.) in DMSO (20 µL) was added, mixed by pipetting and left to incubate for one hour. A column of DMSO only controls, and reagent only controls was also dispensed on the 384-well plate. Following reaction, an aliquot of each reaction mixture (diluted to 2.22 mM) was analysed by LC-MS.

Immediately prior to incubation with proteins (OTUD4, OTUD5, and OTUD7A), each reaction mixture was quenched with hydroxylamine (100 µM). The quench was not performed with OTUD7B or ZRANB1 due to destabilisation of the proteins.

1 µM OTUD4, OTUD5 or 0.5 µM OTUD7A, OTUD7B, ZRANB1 were incubated with 50 µM HTC-D2B library for 24 h at 4 °C, in 25 mM HEPES pH 7.5, 50 mM NaCl buffer. Intact protein LC-MS was performed as described above.

### Compound resynthesis

Compounds **11–30** were purchased from Enamine. Internal compound quality control upon arrival of commercial order was carried out by LC-MS, and compound purity by LC-MS are reported in Supplementary Data 5.

### Kinetic characterisation

Recombinant OTUD7B and OTUD7A were both characterised against compounds **29** and **30**. A dilution series in DMSO was prepared for the compound, and 1 µL added to three separate wells in a 384-well plate, representing technical triplicates of each condition. 99 µL of 0.5 µM OTUD7B or OTUD7A in 25 mM HEPES pH 7.5, 50 mM NaCl buffer was added to the wells and mixed thoroughly (final compound concentrations 100, 50, 25, 12.5, 6.25, 3.125, 1.56 µM). This mixture was then dispensed into

8 wells of 10 μL each in a new 384-well plate, one for each time point. The plate was incubated at 4 °C during intact protein LC-MS. Intact protein LC-MS was performed as previously described above, at approximate time points 0, 1, 2, 4, 6, 8, and 12 h, and deconvoluted as above. The exact times of each measurement were saved with each reading and used for kinetics calculations. Labelling percentages were plotted against time in GraphPad Prism v.10, and curves fitted separately for each replicate using one-phase association, with constraints Y0 = 0 and plateau = highest labelling percentage. Rate constants ($k_{obs}$, generated in GraphPad Prism as K values) were then plotted against concentration in triplicate, and straight lines fitted with constraint Yintercept = 0. Data are presented as mean ± SD, $n = 3$. Slope values were converted from $μM^{-1} h^{-1}$ to $M^{-1} s^{-1}$ to give $k_{inact}/K_I$ values. For OTUD7B kinetics with compound **29**, 100 μM $k_{obs}$ value was not used to calculate $k_{inact}/K_I$, as it was outside the linear range. Errors are reported as standard error, as calculated in GraphPad Prism v.10.

## Biochemical assays

DUB activity assays were performed as previously published[37].

Prior to inhibition assays, optimal conditions for each DUB were found by performing plate-based fluorescence time courses with a matrix of DUB and substrate dilutions. All assays were performed in 50 mM HEPES pH 7.5, 100 mM NaCl, 1 mM EDTA, 0.05% Tween20, 10 mM DTT. The reactions were initiated with the addition of Ub-Rho110Gly and monitored by measuring the rhodamine fluorescence every 30 s at room temperature in a Clariostar Plus plate reader. Conditions were chosen so that the rate of reaction was within the linear phase.

The following concentrations were used: 100 nM Ub-Rho110Gly (UbiQ; UbiQ-002) with 12.5 nM OTUD7B; 200 nM Ub-Rho110Gly with 12.5 nM OTUD7A. DUBs (2.5 μM) were pre-treated with chosen fragments (0–200 μM) for 3 h at room temperature. Following dilution of the DUBs, reactions were initiated with the addition of Ub-Rho110Gly and monitored as above. Initial reaction rates were calculated by taking away background fluorescence of Ub-Rho110Gly hydrolysis and plotting fluorescence against time. Reaction rates were normalised against samples treated with the lowest concentration of fragment and plotted as percentages of residual DUB activity against fragment concentration using GraphPad Prism (v. 9). The curves were fitted using four parameter nonlinear regression with constraints bottom = 0, top = 100. Each experiment was set up as technical triplicates. Outliers were removed before curve fitting. One replicate was removed as an outlier from the concentration-response curve of compound **28** with OTUD7A due to a technical fault with the plate reader.

## SiteID experiments

SiteID experiments were performed as previously described[37].

Recombinantly expressed and purified OTUD7B aa129–538 (10 μM) was incubated with compound **29** (50 μM) for 20 h at 4 °C, in 50 mM HEPES pH 7.5, 150 mM NaCl buffer. The protein–ligand complex was purified by size exclusion chromatography, and stored in 50 mM HEPES pH 7.5, 150 mM NaCl, 0.5 mM TCEP.

In total 5 μL of 20% SDS was added to 20 μL of labelled protein sample and sample was vortexed briefly. The sample was reduced and alkylated by subsequent addition and 30 min incubation at room temperature (600 rpm) of 2 μL of 250 mM DTT (20 mM final concentration) and 2 μL of 500 mM iodoacetamide (40 mM final concentration). The sample was acidified by adding 3 μL of 27.5% phosphoric acid (2.5% final concentration) followed by addition of 165 μL of S-Trap binding buffer (100 mM TEAB, 90% MeOH, pH 8.5). The sample was vortexed briefly and then loaded onto a S-Trap micro column followed by centrifugation (4000 rcf, 30 s). The captured proteins were washed three times with 150 μL of S-Trap binding buffer (100 mM TEAB in 90% MeOH, pH 8.5). The column was centrifuged (4000 rcf, 30 s) between the washes.

20 μL of trypsin (0.05 μg/μL) in S-Trap digestion buffer (50 mM TEAB, pH 8.5) was added to the column and sample was incubated at 47 °C for 2 h. Peptides were eluted from S-Trap column by subsequent addition and centrifugation (4000 rcf, 30 s) of 40 μL of digestion buffer (50 mM TEAB,

pH 8.5), 40 μL of 0.1% formic acid in water and 40 μL of 0.1% formic acid in 50% acetonitrile. The sample was dried overnight in Labconco CentriVap Benchtop Vacuum Concentrator at 4 °C.

Dried peptides were resuspended in 0.1% trifluoroacetic acid to a final concentration of 0.5 μg/μL and 3 μL was injected onto an Ultimate 3000 RSLCnano system (Thermo Fisher Scientific) coupled with an Orbitrap Fusion Lumos Tribrid mass spectrometer (Thermo Fisher Scientific). Peptide sample was separated with a C18 PepMap trap column (2 cm × 100 μm ID, PepMap C18, 5 μm particles, 100 Å pore size, Thermo Fisher Scientific) followed by a 50 cm EASY-Spray column (50 cm × 75 μm ID, PepMap C18, 2-μm particles, 100 Å pore size, Thermo Fisher Scientific). Buffer A was 0.1% formic acid, 5% DMSO and Buffer B was 75% acetonitrile, 0.1% formic acid, 5% DMSO. Peptides were separated by holding 2% Buffer B for 6 min before a step to 8% Buffer B in 1 min. Peptides were then separated with a linear gradient of 8–45% (Buffer B) over 40 min followed by a step from 45 to 95% Buffer B in 2 min at 250 nL/min and held at 95% for 5.5 min. The gradient was then decreased to 2% Buffer B in 0.5 min at 250 nL/min for 15 min.

The data was acquired in "TopSpeed" data dependant mode in positive ion mode with a 3 s cycle time. Full scan spectra were acquired over the mass range 375–1500 *m/z* at a resolution of 120,000 (at 200 *m/z*), with an automatic gain control (AGC) target of standard and a maximum injection time set to auto. Charge states included for analysis were $2–5^{+}$ ions. For MS/MS analysis, the Orbitrap resolution was set to 50000 (at 200 *m/z*). A minimum AGC target was set to standard and the most intense precursor ions were isolated with a quadrupole mass filter width of 1.2. Precursors were subjected to higher-energy collisional dissociation (HCD) fragmentation that was performed using a stepped-step collision energy of 25, 29, and 32%.

Raw files were searched against the Swissprot database using Mascot software (Matrix Science, Version 2.8.0). A 10 ppm mass tolerance for peptides and 0.5 Da mass tolerance for fragment ions were selected. Trypsin was selected as the enzyme and up to 2 missed cleavages were allowed. Carbamidomethylation (C) and oxidation (M) were included as variable modifications. Compound specific modification was added to the database and allowed on cysteine residues only and added to search as variable modification. Validation of binding was performed with manual interpretation of raw spectra.

## Docking experiments

OTUD7B OTU domain crystal structure (PDB 5LRU[24]) was imported into Molecular Operating Environment 2020.0901 (Chemical Computing Group, Montreal, Canada), and prepared using the in-built 'Quick-Prep' function (default parameters). The covalent docking protocol implemented in MOE was employed to generate docking conformations of compounds **29** and **30**, attached to active site Cys194 using the 'alpha_halocarbonyl_S' reaction. Refinement was carried out using the rigid receptor method, based on the GBVI/WSA dG scoring functionality, to give five final poses. The highest scoring pose identified by the docking was taken forward for further molecular analysis. Generation of a 2D ligand interactions map for the highest scoring docking pose was also performed within Molecular Operating Environment 2020.0901 (Chemical Computing Group, Montreal, Canada), using the 'Ligand Interactions' function. Figures of docked ligands were generated in PyMOL 2.3.1 (Schrödinger, LLC), using protein in surface representation, and ligands in stick or sphere representation.

## Reporting summary

Further information on research design is available in the Nature Portfolio Reporting Summary linked to this article.

## Data availability

The mass spectrometry proteomics data have been deposited to the ProteomeXchange Consortium (http://proteomecentral.proteomexchange. org) via the PRIDE partner repository[52] with the dataset identifier

PXD054883 and PXD057851. Additional supplementary information is provided in Supplementary Information and Supplementary Data files 1–8.

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

## Acknowledgements

We would like to acknowledge Harry Wilders and Sam Rowe for HTC-D2B platform development, and Emma Grant for platform maintenance. We would also like to acknowledge Emma Cawood for IA-DTB Python analysis script development and Paula Magarinos for assistance with the data analysis. In addition, we would like to thank both Cell Services and the Proteomics STP, and the Francis Crick Institute for their support with this manuscript. This work was supported by the Francis Crick Institute, which receives its core funding from Cancer Research United Kingdom (CC 2075), the United Kingdom Medical Research Council (CC 2075), and the Wellcome Trust (CC 2075); by the Biotechnology and Biological Research Council, BB/T014547/1 to K.R. and D.H.; and by the Engineering and Physical Sciences Research Council, EP/V038028/1 to D.H., J.T.B., and K.R. For the purpose of open access, the author has applied a CC BY public copyright license to any author-accepted manuscript version arising from this submission. Figs. 1A and 3A were made using Biorender.com under an institutional license belonging to The Francis Crick Institute.

## Author contributions

A.V., C.R.K., and K.A.M. performed all experiments and wrote the manuscript. A.V., C.R.K., and K.A.M. contributed equally to this manuscript. J.P. carried out library management, and W.M. performed library annotation and IA-DTB chemoproteomics. J.M.S., D.H., J.B., and K.R. contributed to data analysis and discussion. J.B. and K.R. jointly supervised the study. All authors read and edited the final manuscript.

## Funding

## Competing interests

The authors declare no competing interests.
