## [Transparent Peer Review file · Communications Chemistry]

Enantioselective OTUD7B fragment discovery through chemoproteomics screening and high-throughput optimisation

Corresponding Author: Dr Katrin Rittinger

Version 0:

Reviewer comments:

Reviewer #1

(Remarks to the Author)

The manuscript by Vuorinen, Kennedy, McPhie et al. from the Bush and Rittinger groups describe an improved version of their previously reported chemoproteomics platform and the identification of covalent fragments binding to OTU deubiquitinases. The identification of novel and specific chemical matter for their targeting is in high demand due to the emerging therapeutic prospects of DUB inhibitors as well as DUB-recruiting bifunctional small molecules. The described platform uses different high-end mass spectrometry approaches (both for chemoproteomics as well as intact protein analysis, with notable improvements over the previous version, trading some coverage for speed) combined with high throughput chemistry, and thereby pushes the boundaries of the field. These approaches are well described and well-illustrated in the manuscript. The identification and initial characterization of an enantioselective fragment for OTUD7B validates the approach.

However, the manuscript suffers from several issues which preclude publication in its current form. These include missing chemical synthesis/characterization data, poor reference to the literature, technical inconsistencies rendering some data uninterpretable, some overstatements, and missing validation of the binding mode. In addition, the high degree of automated/processed data analysis makes it difficult to assess raw data quality. These points are described here in detail and should be addressed before publication can be considered:

- The authors mention in the text "Compounds 11 – 28 were successfully resynthesized, purified and taken forward". However, the manuscript lacks any descriptions of syntheses or chemical characterization data (1H, 13C, MS). In line with journal standards, these data must be added, including ee% for compounds 29/30 or a reference to the ee of their starting materials. Fig. S4 states that some compounds could not be made. If so, please rephrase the text accordingly.
- Regarding the high throughput chemistry, the authors mention that understandably „product purities were not uniform across the plate“. It would be helpful if the authors could include a few exemplary LC-MS traces to demonstrate the properties of the material that was screened. These will not be completely pure compounds, but this information might be useful to others adopting the approach.
- Also regarding the HTC-D2B assay, the text states "Following analysis by LC-MS to determine conversion rates⁴¹, and a reagent quench with hydroxylamine" but the methods state that there was no quench performed for OTUD7B and ZRANB1. Why not? Fig. 3B shows these two DUBs have massively elevated hit rates, which the authors relate to their fitting selection of amines in the second library, when in fact (if there was no quench for some DUBs) this may be a technical artefact due to retained chemical reactivity. This may be particularly relevant for OTUD7B which has a very high propensity to react with electrophiles.
- The degree of labeling in this intact protein mass assay is used also to compare across different DUBs, but rarely 100% labeling is reached even at the high 200 μ M concentrations. Likewise, OTUD5 covalent labeling seems to be leveling off at around 60% (OTUD7B at around 90%), so there be some heterogeneity in protein batches which should be considered (maybe with a Ub-VS measurement, and normalization to the highest labeling observed there as the portion of catalytically competent enzyme). Otherwise comparisons between DUBs and regarding specificity have to be treated with caution.
- The high degree of processed data makes it difficult to assess the mass spec data quality. Please include a few deconvoluted mass spectra of different DUBs and fragments to illustrate the results. The assay seems to have been carried out at 4°C – how does the reactivity change when the incubation is done at more relevant temperatures (e.g. RT or 37°C). This could be combined in a new figure, as this will demonstrate the utility of the fragments and may remove a bias towards the DUB with the highest remaining reactivity at 4°C.

- The same point regarding the processed data can be made for the chemoproteomics. This reviewer understands the needs to compress the data into heatmaps, but 1-2 volcano plots should be included to (e.g. for the most advanced compounds) to allow readers to assess the data, also taking statistical significance of the results into account. Please include statements on the number of replicates for all mass spec experiments.
- In Fig. 2A the scale starts at 1 (should actually be minus 1?), meaning that a fragment can inhibit a DUB up to 50% and the field is still white. This offset may make the data look cleaner, but gives a false impression of selectivity. The same is used for statements like „Did not label OTUD7A“ in relation to Fig. 2C, when several compounds label up to 40%.
- The data in Supp Fig. 3. e.g. on fragment 3 look very strong, and overall these are great data to build from. However, the authors should mention explicitly that fragment 2 was used in their previous publication (Cookson et al.), was termed fragment 26 there and was already there found to be highly OTUD7B reactivity (Fig. 5A+B).
- OTUD7B is also named Cezanne 1, and Mission Therapeutics filed patents for highly potent, covalent Cezanne 1 inhibitors in 2015-16 (e.g. EP4067355A1, EP3394049B1). The latter includes an enantioselective pair of compounds 12: IC50 < 0.1 uM; 13: IC50 1-20 uM, which mirrors the about 20x difference the authors observe for their compound. The compounds also have the same difference between stereocenter and electrophilic carbon. This should be included, and any statements like “first published chemical structure of an OTUD7B inhibitor“ be adjusted.
- Likewise, the authors should reference recent and relevant work from the Geurink (HT chemistry for covalent probes, UCHL1 compounds in live cells), Gersch (chemogenomic probe pairs, specificity analysis), Flaherty (UCHL1 covalent fragment, fragment hit expansion) labs.
- These references will also be helpful to put the obtained fragments into context. The authors use the word “probe” in the title (which should be avoided, as compound 29 is not nearly as specific or characterized to justify this designation). In the absence of a specificity or phenotype analysis in live cells (as done by all these groups for UCHL1 targeting probes / tool compounds), even “tool compound” / “biological tools” is a bit of stretch. These limitations should be addressed or openly discussed.
- The curves for OTUD7A in Supp Fig. 6 look odd. As the specificity of the enantioselective fragment for OTUD7B is based on these data, can the authors comment on how confident they are on these data? The focus on abs. log2 ratios for data visualization makes sense, but when it comes to specificity, this representation is misleading (e.g. Fig. 4B: -2.5 means 83% labeled, -5.5 means 98% labeled; this looks like a big difference, but is not different enough for a cellular experiment).
- The docking figures at the end capturing the bound state lack validation. The authors should attempt to validate the obtained poses by mutation. Firstly, by assessing that actually the catalytic cysteine is bound, and then secondly, e.g. by mutating Asp191 or Leu294 to assess whether these residues contribute to enantioselectivity as would be predicted by the shown geometries. A chemical validation (fragments with substituents at the phenyl ring should increase discrimination between stereoisomers) will likely be out of scope, but at the moment this seems very preliminary (not even considering that docking the “post-reaction” state may not always capture what drives reaction speed). The catalytic cysteine may seem obvious but other fragments/compounds bind non-catalytic Cys in e.g. OTUB1 or UCHL1. Providing these data is important to justify the claim that the compound provides an “excellent starting point for further structure-based optimization”.

Minor points:

- Supp Fig. 3C: The previous finding that OTUD5 phosphorylation mediates ubiquitin recognition (i.e. not nucleophilicity) could be added as this is in line with these results.
- The rationale for therapeutic targeting of OTU DUBs is oddly derived from their ability to cleave specific polyUb linkages without any further information, this should be rephrased or made stronger by direct disease relation. (Refs 17 and 18 include an X).
- „Only a handful of OTU DUBs have high-quality chemical tools“ requires a citation. I would also doubt that this statement is accurate.
- „fragment hits against this family were in an optimal reactivity space“ What does this mean?
- In Supp Fig. 2, why are there fewer fragments (fewer columns) for A (ATXN3) and C (UBP DUBs) than in B (UCHs)?
- IC50 values of covalent compounds heavily depend on the experimental conditions, and while frequently used, should be treated with care. Kobs/l values should be recorded at least for fragment 29+30 to allow cross-comparison to other studies.
- Supp Fig. 6A shows an IC50 of 1.6 uM and seems to relate to data shown in Fig. 4A, however, the data curves there suggest the 50% transition to occur more at 3.5 uM (was the upper 100% boundary fixed during the curve fitting)? This would also be consistent with the in vitro IC50, and the 20x reactivity difference.

Reviewer #2

(Remarks to the Author)

This manuscript by Vuorinen and coauthors described a chemoproteomics screening of a small library of chloroacetamide fragments targeting DUBs and the subsequent optimization of the hits through high-throughput chemistry and cell lysate-based screening using improved LC-MS workflow. This work demonstrated improved screening capacity of the cell lysate-based chemoproteomics workflow, which is a major strength of the manuscript. Further, an interesting enantioselective compound was identified with demonstrated selectivity toward OTUD7B. Although compound 29 is not yet ready to be used as a tool compound for OTUD7B, it provides a promising starting point for future scaffold expansion and optimization. Overall this is a well-written manuscript and will be of high interest to the DUB field. Below are points that should be addressed before the publication of this work.

The authors stated that “all fragments which targeted OTUD7A by chemoproteomics did not label OTUD7A by intact protein LC-MS”. It does not seem to be an accurate description of the results shown in Figure 2C regarding the OTUD7A in vitro and chemoproteomics data. For example, fragment 5 and 7 seem to label OTUD7A efficiently in both experiments, while fragment 1 showed good in vitro labeling but poor chemoproteomics labeling. Please clarify.

The authors should include an SI figure similar to that in Figure 2A for other families of DUBs. This information would be useful to the readers and give this work broader significance.

The authors need to demonstrate that compounds 29 and 30 label the catalytic cysteine of OTUD7B using purified DUB and trypsin-digestion MS analysis.

It is not entirely clear how the percentage quantification (such as those in Figure 4D & E) of Biotin-Ahx-Ub-VS labeling of DUBs in the cell lysates was done. Were LFQ values used to calculate the percentage of DUB being captured and detected by mass spectrometry? Please discuss this in the Methods section.

One caveat of the above approach is that the Ub-VS probe may label different DUBs in the lysate with different affinity/potency. Thus the apparent IC50 obtained by fitting the curves in Figure 4D & E may not reflect accurately the potency of the compound. A more rigorous kinact/KI determination would be beneficial to confirm the selectivity of the compounds versus selected DUBs.

The authors did not discuss the off-target labeling of proteins other than DUBs by the chloroactamide compounds. This is an important consideration in developing covalent inhibitors targeting cysteine.

In the HTC steps, the amino coupling reaction was quenched with hydroxylamine for OTUD4, OTUD5 and OTUD7A but not for OTUD7B or ZRANB1. Why?

The comparison of the docking of compound 29 and 30 to OTUD7B was simply based on the H-bonds. However, an equal number of H-bonds are found in both cases. It is thus unlikely the major factor of the selectivity. How different are the energetics of the two compounds binding to the OTUD7B active site? How about a steric clash? Figure 5B & C can include a space-filling model of the compounds.

Reviewer #3

(Remarks to the Author)

Please see attached comments

Version 1:

Reviewer comments:

Reviewer #1

(Remarks to the Author)

The authors have done a great job in thoroughly revising their manuscript according to all comments of the three reviewers with additional data, additional data representations, clarifications, and careful interpretations of their data. These changes have further improved the quality of the manuscript, which - as stated in the initial report - pushes the boundaries of the field. I am thus happily supporting publication in Communications Chemistry and congratulate the authors on their contribution to the field!

Reviewer #3

(Remarks to the Author)

Following comprehensive response to reviewer comments, publication is recommended.

We wish to thank the reviewers for their supportive comments and helpful suggestions for improving our manuscript. We agree with the majority of the reviewers' comments, and we have performed several new experiments and adapted our manuscript to address their points. We believe the revised manuscript is much improved as a result of their suggestions.

For clarity, we have addressed and responded to each point individually, in purple, below.

Points from Reviewer 1:

The manuscript by Vuorinen, Kennedy, McPhie et al. from the Bush and Rittinger groups describe an improved version of their previously reported chemoproteomics platform and the identification of covalent fragments binding to OTU deubiquitinases. The identification of novel and specific chemical matter for their targeting is in high demand due to the emerging therapeutic prospects of DUB inhibitors as well as DUB-recruiting bifunctional small molecules. The described platform uses different high-end mass spectrometry approaches (both for chemoproteomics as well as intact protein analysis, with notable improvements over the previous version, trading some coverage for speed) combined with high throughput chemistry, and thereby pushes the boundaries of the field. These approaches are well described and well-illustrated in the manuscript. The identification and initial characterization of an enantioselective fragment for OTUD7B validates the approach.

We thank reviewer 1 for their encouraging comments.

However, the manuscript suffers from several issues which preclude publication in its current form. These include missing chemical synthesis/characterization data, poor reference to the literature, technical inconsistencies rendering some data uninterpretable, some overstatements, and missing validation of the binding mode. In addition, the high degree of automated/processed data analysis makes it difficult to assess raw data quality. These points are described here in detail and should be addressed before publication can be considered:

We acknowledge the reviewer's comments and have addressed the detailed comments below.

- The authors mention in the text "Compounds 11 – 28 were successfully resynthesized, purified and taken forward". However, the manuscript lacks any descriptions of syntheses or chemical characterization data (1H, 13C, MS). In line with journal standards, these data must be added, including ee% for compounds 29/30 or a reference to the ee of their starting materials. Fig. S4 states that some compounds could not be made. If so, please rephrase the text accordingly.

We thank reviewer 1 for highlighting that our wording was unclear. All compounds used were purchased from Enamine (catalogue numbers are in Supplementary Table 1). Upon arrival, compounds were quality checked by LC-MS, and these QC purity values have been added into Supplementary Table 1. We have altered the text on p.11, line 5 in the main text to 'purchased as purified compounds', and p.24, lines 23-27 in the

methods section to 'Compounds 11 – 30 were purchased from Enamine. Internal compound quality control upon arrival of commercial order was carried out by LC-MS, and compound purity by LC-MS are reported in Supplementary Table 1.' to make this clear.

- Regarding the high throughput chemistry, the authors mention that understandably „product purities were not uniform across the plate“. It would be helpful if the authors could include a few exemplary LC-MS traces to demonstrate the properties of the material that was screened. These will not be completely pure compounds, but this information might be useful to others adopting the approach.

We have added Supplementary Figures 7A and 7B to demonstrate the distribution of conversion rates with HTC, and examples of LC-MS traces for each category of conversion (ie. 0%, <25%, 25-50%, >50% conversion).

- Also regarding the HTC-D2B assay, the text states “Following analysis by LC-MS to determine conversion rates⁴¹, and a reagent quench with hydroxylamine“ but the methods state that there was no quench performed for OTUD7B and ZRANB1. Why not? Fig. 3B shows these two DUBs have massively elevated hit rates, which the authors relate to their fitting selection of amines in the second library, when in fact (if there was no quench for some DUBs) this may be a technical artefact due to retained chemical reactivity. This may be particularly relevant for OTUD7B which has a very high propensity to react with electrophiles.

Thank you to reviewer 1 for raising this important point. We did not use a hydroxylamine quench for OTUD7B or ZRANB1 because both proteins were unstable to the hydroxylamine treatment, as observed by significant differences in the intact protein LC-MS traces. In contrast, the other DUBs tolerated the quench conditions well. It is possible that this affected the number of hits for OTUD7B and ZRANB1. Because of this, we were careful not to make comparisons of hit compounds from the HTC plates across the DUBs, only to contextualise hits for a given DUB. To clarify this, we have added the words 'where tolerated' on p.10, line 20, and the sentence 'OTUD4, OTUD5 and OTUD7A tolerated exposure to hydroxylamine well, however we observed a loss of MS signal for OTUD7B and ZRANB1, and screening of the HTC library was repeated against these two proteins without the reagent quench step' on p.10, lines 24-26. Additionally, we have added the sentence 'To avoid comparisons between labelling events with and without the hydroxylamine quench step, we focused on compounds that showed improved labelling for a given protein at this stage, rather than comparing fragment labelling across all five proteins.' on p.10, lines 31-34.

- The degree of labeling in this intact protein mass assay is used also to compare across different DUBs, but rarely 100% labeling is reached even at the high 200 uM concentrations. Likewise, OTUD5 covalent labeling seems to be leveling off at around 60% (OTUD7B at around 90%), so there be some heterogeneity in protein batches which should be considered (maybe with a Ub-VS measurement, and normalization to the highest labeling observed there as the portion of catalytically competent enzyme). Otherwise comparisons between DUBs and regarding specificity have to be treated with caution.

Thank you for raising this point. There are unavoidable variabilities in heterogeneity across different recombinant DUB preps. As with the point above, we have avoided making comparisons of hits across different DUB labelling with our recombinant proteins, and have instead focused on comparison of hits for one given DUB. To clarify

this, we have added the sentence 'We observed that different DUBs reached differential maximum labelling percentages (Supplementary Figure 5A and B), suggesting heterogeneity between protein batches, and therefore direct comparisons between labelling percentages across DUBs were treated with caution' on p.8, lines 20-23.

- The high degree of processed data makes it difficult to assess the mass spec data quality. Please include a few deconvoluted mass spectra of different DUBs and fragments to illustrate the results.

Thank you for raising this point. We have added examples of deconvoluted spectra for our in vitro validation (Supplementary Figure 6) and for our HTC-D2B screening (Supplementary Figure 7C) for each DUB.

- The assay seems to have been carried out at 4°C – how does the reactivity change when the incubation is done at more relevant temperatures (e.g. RT or 37°C). This could be combined in a new figure, as this will demonstrate the utility of the fragments and may remove a bias towards the DUB with the highest remaining reactivity at 4°C. We have performed labelling experiments with recombinant proteins at 4°C in order to maximise protein stability over the 24-hour incubation period. We believe that screening at higher temperatures over a 24-hour incubation period would risk partial denaturation/precipitation of protein and hence affect the reproducibility of our experiments.

- The same point regarding the processed data can be made for the chemoproteomics. This reviewer understands the needs to compress the data into heatmaps, but 1-2 volcano plots should be included to (e.g. for the most advanced compounds) to allow readers to assess the data, also taking statistical significance of the results into account. Please include statements on the number of replicates for all mass spec experiments.

Thank you for this comment. We have added volcano plots for our chemoproteomics experiments with compounds 1 – 7 (Supplementary Figure 4) and compounds 29 and 30 (Supplementary Figure 10C) to address this point. We have also added the number of replicates (n=3 for UbVS chemoproteomics, and n=4 for IA-DTB chemoproteomics) to all relevant figure captions and in the Methods section.

- In Fig. 2A the scale starts at 1 (should actually be minus 1?), meaning that a fragment can inhibit a DUB up to 50% and the field is still white. This offset may make the data look cleaner, but gives a false impression of selectivity. The same is used for statements like „Did not label OTUD7A“ in relation to Fig. 2C, when several compounds label up to 40%.

This is a very good point. We have amended the scales for all heatmaps accordingly to start at 0 (Figure 2A, and Supplementary Figures 2 and 3), and have highlighted our use of absolute average log₂ values for clarity of visualisation in the figure captions. We have also changed the RHS y-axis on Figure 2C to represent chemoproteomics percentage labelling as '100-Biotin-Ahx-Ub-VS probe labelling', which we think improves the visualisation of data. We have also amended the statement 'did not label 7A' to 'For OTUD7A, chemoproteomics and intact protein LC-MS were less concordant, with some discrepancies between the two techniques observed for several fragments labelling OTUD7A' on p.8, lines 17-20.

- The data in Supp Fig. 3. e.g. on fragment 3 look very strong, and overall these are great data to build from. However, the authors should mention explicitly that fragment 2 was used in their previous publication (Cookson et al.), was termed fragment 26 there and was already there found to be highly OTUD7B reactivity (Fig. 5A+B).

Thank you for highlighting this. We have added the sentence 'The selected fragments included compound **2**, which was profiled as an OTUD7B inhibitor in our previous work³⁷ (therein compound **26**).' to p.8, lines 10-12, to draw attention to our previous work in more detail.

- OTUD7B is also named Cezanne 1, and Mission Therapeutics filed patents for highly potent, covalent Cezanne 1 inhibitors in 2015-16 (e.g. EP4067355A1, EP3394049B1). The latter includes an enantioselective pair of compounds 12: IC₅₀ < 0.1 uM; 13: IC₅₀ 1-20 uM, which mirrors the about 20x difference the authors observe for their compound. The compounds also have the same difference between stereocenter and electrophilic carbon. This should be included, and any statements like "first published chemical structure of an OTUD7B inhibitor" be adjusted.

Thank you for drawing attention to this oversight on our part. We have removed this claim from our manuscript and added the following sentences: to the introduction on p.2, lines 29-30 'A series of cyanopyrrolidine inhibitors of OTUD7B have also been disclosed in patents from Mission Therapeutics...'; and to the results and discussion on p.14, lines 20-22 'Interestingly, this observation concurs with enantioselectivity observed for a pair of published cyanopyrrolidine OTUD7B inhibitors¹⁹.'

- Likewise, the authors should reference recent and relevant work from the Geurink (HT chemistry for covalent probes, UCHL1 compounds in live cells), Gersch (chemogenomic probe pairs, specificity analysis), Flaherty (UCHL1 covalent fragment, fragment hit expansion) labs.

We have added references (numbers 13 – 15) to our manuscript, described in the sentence on p.2, lines 22-26 'In fact, UCHL1 has been the target of several successful covalent chemical probe discovery campaigns, where optimised cyanopyrrolidine and chloroacetamide activity-based probes have been used to characterise the mechanisms of deubiquitination and the downstream effects of UCHL1 inhibition in live cells and zebrafish embryos.'

- These references will also be helpful to put the obtained fragments into context. The authors use the word "probe" in the title (which should be avoided, as compound 29 is not nearly as specific or characterized to justify this designation). In the absence of a specificity or phenotype analysis in live cells (as done by all these groups for UCHL1 targeting probes / tool compounds), even "tool compound" / "biological tools" is a bit of stretch. These limitations should be addressed or openly discussed.

Thank you for this highlighting this point. We have changed the word 'probe' to 'fragment' in the title and abstract, and we have added the following sentence to the conclusion on p.18, lines 26-27 'a novel pharmacophore and the first use of a chloroacetamide electrophile for OTUD7B inhibition.'

- The curves for OTUD7A in Supp Fig. 6 look odd. As the specificity of the enantioselective fragment for OTUD7B is based on these data, can the authors comment on how confident they are on these data? The focus on abs. log₂ ratios for data visualization makes sense, but when it comes to specificity, this representation is

misleading (e.g. Fig. 4B: -2.5 means 83% labeled, -5.5 means 98% labeled; this looks like a big difference, but is not different enough for a cellular experiment).

With regards to the curves for OTUD7A in the now renumbered Supplementary Figure 10B, we agree that these are not ideal IC50 curves. However, the chemoproteomics data for OTUD7B gave robust concentration-response curves and small error bars per point, so we are confident in the OTUD7B data. We have added a sentence into the main text to address the OTUD7A chemoproteomics on p.12, lines 26-29 'Due to identification of a small number of low intensity peptides for OTUD7A, we were unable to measure accurate IC50 values for OTUD7A by chemoproteomics, and therefore could not compare compound selectivity between OTUD7B and OTUD7A by chemoproteomics.' We have instead addressed the question of selectivity over OTUD7A using biochemical assays (Figure 5A and Supplementary Figure 11A) and in vitro kinetics characterisation (Supplementary Figure 11B – this is an additional experiment performed to address reviewer 1 minor point, and reviewer 2 and 3 comments, see below for details) with enantiomeric compounds 29 and 30, which confirmed the specificity of these compounds for OTUD7B over OTUD7A.

With regards to the use of abs log2 ratios in Figure 4B (the 200 and 50 uM), this is a typical and broadly adopted method for displaying of proteomics data, particularly when analysing at a single concentration. However, we have also added in another representation of this data in the form of a volcano plot (Supplementary Figure 10C), which also allows visualisation of specificity in a different way. We agree with the reviewer that log2 scale is a crude method for performing a detailed comparison of potencies of compounds. It is for this reason that we ran selected compounds in full concentration response, and this data we have indeed represented as percentage labelling, in agreement with the reviewers suggestion, in Figure 4D and E (and Supplementary Figure 10B).

- The docking figures at the end capturing the bound state lack validation. The authors should attempt to validate the obtained poses by mutation. Firstly, by assessing that actually the catalytic cysteine is bound, and then secondly, e.g. by mutating Asp191 or Leu294 to assess whether these residues contribute to enantioselectivity as would be predicted by the shown geometries. A chemical validation (fragments with substituents at the phenyl ring should increase discrimination between stereoisomers) will likely be out of scope, but at the moment this is seems very preliminary (not even considering that docking the "post-reaction" state may not always capture what drives reaction speed). The catalytic cysteine may seem obvious but other fragments/compounds bind non-catalytic Cys in e.g. OTUB1 or UCHL1. Providing these data is important to justify the claim that the compound provides an "excellent starting point for further structure-based optimization".

Thank you for this comment. To validate the site of binding, ie the residue at which the chloroacetamide binds, we performed an additional site identification experiment using fragment labelled purified DUB. After trypsin digestion the peptides were analysed by LC/MS-MS (added into Figure 5D), which confirmed the site of modification by compound 29 is the catalytic cysteine residue, Cys194. This validates the residue used for covalent docking, however we have also put less emphasis on these docking experiments by moving the figures into Supplementary Figure 12 and adding the sentence on p.17, lines 29-32 'As such, although this covalent molecular docking suggests a possible binding mode, structural biology would be required to further elucidate compound mechanism of action and explain the differences in potency observed between the two enantiomers.' We have also removed the words 'structure-

based' from the sentence 'provides an excellent starting point for further structure-based optimization' on p.17, line 37. We believe that extensive mutagenesis and further SAR exploration is out of scope.

Minor points:

Thank you for very much for highlighting these minor points. We have addressed these points in the manuscript as outlined below.

- Supp Fig. 3C: The previous finding that OTUD5 phosphorylation mediates ubiquitin recognition (i.e. not nucleophilicity) could be added as this is in line with these results. We have added the sentences 'OTUD5 is known to be activated in cells through phosphorylation, altering its propensity to bind ubiquitin. To investigate whether we had selective fragment binders for either form of OTUD5, we validated both the inactive and activated forms, however we observed similar labelling for both states (Supplementary Figure 5C), supporting that OTUD5 phosphorylation does not alter the nucleophilicity of the catalytic cysteine.' on p.8, lines 28-33.

- The rationale for therapeutic targeting of OTU DUBs is oddly derived from their ability to cleave specific polyUb linkages without any further information, this should be rephrased or made stronger by direct disease relation. (Refs 17 and 18 include an X). The X in references 17 and 18 (now references 22 and 23) has been removed. We have also rephrased the paragraph you describe on p.3, lines 9-15, to improve the rationale for therapeutic targeting.

- „Only a handful of OTU DUBs have high-quality chemical tools“ requires a citation. I would also doubt that this statement is accurate.

We have rephrased this statement to say 'many members of the OTU DUBs have yet to be targeted by well-characterised chemical tools' on p.6, lines 2-3.

- „fragment hits against this family were in an optimal reactivity space“ What does this mean?

We have rephrased this statement to say 'We observed that some of these fragment hits labelled one OTU DUB selectively, while other fragments labelled several OTU DUBs with minimal labelling of other DUB families. Crucially, none of the fragment hits against OTU DUBs labelled more than 15 other non-OTU DUB family proteins, which suggested good protein ligandability without extensive fragment promiscuity (Supplementary Figure 3A, B and C).' on p.6, lines 6-10.

- In Supp Fig. 2, why are there fewer fragments (fewer columns) for A (ATXN3) and C (UBP DUBs) than in B (UCHs)?

In Supplementary Figure 2 (now called Supplementary Figure 3), we have only shown the fragment hits which compete at least one DUB in that sub-family – we do not show all of the fragments in the library for ease of visualisation. Thus, there are different numbers of fragment hits identified within each sub-family represented in A) UCH DUB sub-family, B) USP DUB sub-family and C) MJD DUB sub-family. We have made this clearer in the figure caption and with the axis title in the figures. We have also numbered the fragment hits on the x-axis for clarity too. We have also additionally included a new heatmap in Supplementary Figure 2, which shows the entire 227 fragment library against the 43 identified DUBs.

- IC50 values of covalent compounds heavily depend on the experimental conditions, and while frequently used, should be treated with care. Kobs/I values should be recorded at least for fragment 29+30 to allow cross-comparison to other studies.

To complement the IC50 values, we have performed full kinetics characterisation for compounds 29 and 30 against recombinant OTUD7A and OTUD7B (Figure 5B and C, and Supplementary Figure 11B), and we have reported k_{inact}/K_i values for compounds 29 and 30 against OTUD7B (Figure 5C, and in the main text). We were not able to calculate k_{inact}/K_i for compounds 29 and 30 against OTUD7A as the labelling we observed was too low. We describe this kinetic characterisation in the main text on p.15, lines 5-21.

- Supp Fig. 6A shows an IC50 of 1.6 μ M and seems to relate to data shown in Fig. 4A, however, the data curves there suggest the 50% transition to occur more at 3.5 μ M (was the upper 100% boundary fixed during the curve fitting)? This would also be consistent with the in vitro IC50, and the 20x reactivity difference.

The upper boundary was unconstrained in the original version of our manuscript, we have therefore recalculated all IC50 values and TE50 values with the upper boundary now set to top=100 in GraphPad Prism. We have changed the figures and highlighted this in the figure captions (Figure 4D and E, Figure 5A and Supplementary Figures 10B and 11A). We have adjusted the IC50 values in the figures and in the text accordingly. We have also highlighted the curve-fitting method in the methods section on p.21, lines 21-26 to include this change 'Concentration-response curves were fitted using label free protein quantity values. Values were normalised against samples treated with DMSO and plotted as percentages of Biotin-Ahx-Ub-VS labelling against fragment concentration using GraphPad Prism (v. 10). The curves were fitted using four parameter nonlinear regression with constraints bottom=0, top = 100. Each experiment was set up as technical triplicates.' For completeness, we have also reported the 95% confidence interval values as calculated using GraphPad Prism (v. 10), reported in Supplementary Figure 10A and 11A.

Points from Reviewer 2:

This manuscript by Vuorinen and coauthors described a chemoproteomics screening of a small library of chloroacetamide fragments targeting DUBs and the subsequent optimization of the hits through high-throughput chemistry and cell lysate-based screening using improved LC-MS workflow. This work demonstrated improved screening capacity of the cell lysate-based chemoproteomics workflow, which is a major strength of the manuscript. Further, an interesting enantioselective compound was identified with demonstrated selectivity toward OTUD7B. Although compound 29 is not yet ready to be used as a tool compound for OTUD7B, it provides a promising starting point for future scaffold expansion and optimization. Overall this is a well-written manuscript and will be of high interest to the DUB field. Below are points that should be addressed before the publication of this work.

We thank reviewer 2 for their enthusiasm for our work. We have addressed the points raised below.

The authors stated that “all fragments which targeted OTUD7A by chemoproteomics did not label OTUD7A by intact protein LC-MS”. It does not seem to be an accurate description of the results shown in Figure 2C regarding the OTUD7A in vitro and chemoproteomics data. For example, fragment 5 and 7 seem to label OTUD7A efficiently in both experiments, while fragment 1 showed good in vitro labeling but poor chemoproteomics labeling. Please clarify.

We thank this reviewer for raising this important point, which was also raised by reviewer 1. As described in the response to reviewer 1, we have changed the right-side y axis on Figure 2C to represent chemoproteomics percentage labelling as ‘100-Biotin-Ahx-Ub-VS probe labelling’, which we think improves the visualisation and comparison of chemoproteomics and intact protein LC-MS data. We have also amended the statement ‘did not label OTUD7A’ to ‘For OTUD7A, chemoproteomics and intact protein LC-MS were less concordant, with some discrepancies between the two techniques observed for several fragments labelling OTUD7A’ on p.8, lines 17-20.

The authors should include an SI figure similar to that in Figure 2A for other families of DUBs. This information would be useful to the readers and give this work broader significance.

Thank you for this helpful suggestion. In Supplementary Figure 3, we show heatmaps similar to that in Figure 2A, which show the fragment hits which compete at least one DUB in the three other DUB sub-families in A) UCH DUB sub-family, B) USP DUB sub-family and C) MJD DUB sub-family. We have now made this clearer in the figure caption and with the axis title in the figures. We have also numbered the fragment hits on the x-axis for clarity too. We have additionally included a new heatmap in Supplementary Figure 2, which shows the entire 227 fragment library against the 43 identified DUBs.

The authors need to demonstrate that compounds 29 and 30 label the catalytic cysteine of OTUD7B using purified DUB and trypsin-digestion MS analysis.

Thank you for raising this point. We performed an additional site identification experiment using fragment labelled purified DUB. After trypsin digestion the peptides were analysed by LC/MS-MS (added into Figure 5D), which confirmed the site of modification by compound 29 is the catalytic cysteine residue, Cys194. Unfortunately, we could not obtain this information for compound 30, due to the low labelling of compound 30 observed (as exemplified in additional kinetic characterisation experiment shown in Supplementary Figure 11B).

It is not entirely clear how the percentage quantification (such as those in Figure 4D & E) of Biotin-Ahx-Ub-VS labeling of DUBs in the cell lysates was done. Were LFQ values used to calculate the percentage of DUB being captured and detected by mass spectrometry? Please discuss this in the Methods section.

Thank you for highlighting this oversight. We have added a more detailed description of this into the methods section on p.21, lines 21-26.

One caveat of the above approach is that the Ub-VS probe may label different DUBs in the lysate with different affinity/potency. Thus the apparent IC50 obtained by fitting the curves in Figure 4D & E may not reflect accurately the potency of the compound. A more rigorous kinact/KI determination would be beneficial to confirm the selectivity of the compounds versus selected DUBs.

Thank you for raising this important point, which was also raised by reviewer 1. As described in the response to reviewer 1's comment, to complement the IC₅₀ values, we have performed full kinetics characterisation for compounds 29 and 30 against recombinant OTUD7A and OTUD7B (Figure 5B and C, and Supplementary Figure 11B), and we have reported k_{inact}/K_i values for compounds 29 and 30 against OTUD7B (Figure 5C, and in the main text). We were not able to calculate k_{inact}/K_i for compounds 29 and 30 against OTUD7A as the labelling we observed was too low. We describe this kinetic characterisation in the main text on p.15, lines 5-21.

The authors did not discuss the off-target labeling of proteins other than DUBs by the chloroacetamide compounds. This is an important consideration in developing covalent inhibitors targeting cysteine.

Thank you for highlighting this point. To address the proteome-wide selectivity of compounds 29 and 30, we have included an additional chemoproteomics experiment, which uses an iodoacetamide-desthiobiotin (IA-DTB) competitive probe workflow, which we have recently published (biorxiv, <https://doi.org/10.1101/2024.07.25.605137>). A heatmap of compound off-targets is shown in Supplementary Figure 10D, and we have discussed non-DUB off-targets identified in this IA-DTB experiment in the main text on p.14, lines 1-14. We have also included a description of this workflow in the methods section on p.21-23, starting on p.21, line 28. In line with the precedent set in the above recent publication, we have also renamed all chemoproteomics IC₅₀ values to TE₅₀ values (where TE = target engagement).

In the HTC steps, the amino coupling reaction was quenched with hydroxylamine for OTUD4, OTUD5 and OTUD7A but not for OTUD7B or ZRANB1. Why?

Thank you for raising this important point, which was also raised by reviewer 1. As discussed in the response to reviewer 1, we did not use a hydroxylamine quench for OTUD7B or ZRANB1 because both proteins were unstable to the hydroxylamine treatment, as observed by significant differences in the intact protein LC-MS traces. In contrast, the other DUBs tolerated the quench conditions well. It is possible that this affected the number of hits for OTUD7B and ZRANB1. Because of this, we were careful not to make comparisons of hit compounds from the HTC plates across the DUBs, only to contextualise hits for a given DUB. To clarify this, we have added the words 'where tolerated' on p.10, line 20, and the sentence 'OTUD4, OTUD5 and OTUD7A tolerated exposure to hydroxylamine well, however we observed degradation of OTUD7B and ZRANB1, and screening of the HTC library was repeated against these two proteins without the reagent quench step' on p.10, lines 24-26. Additionally, we have added the sentence 'To avoid comparisons between labelling events with and without the hydroxylamine quench step, we focused on compounds that showed improved labelling for a given protein at this stage, rather than comparing fragment labelling across all five proteins.' on p.10, lines 31-34.

The comparison of the docking of compound 29 and 30 to OTUD7B was simply based on the H-bonds. However, an equal number of H-bonds are found in both cases. It is thus unlikely the major factor of the selectivity. How different are the energetics of the two compounds binding to the OTUD7B active site? How about a steric clash? Figure 5B & C can include a space-filling model of the compounds.

We agree with the reviewer's points and have rephrased the discussion in the main text to put less emphasis on these docking experiments. We have moved the docking figures to Supplementary Figure 12 and have included a space-filling model in Supplementary Figure 12C and D. We have removed the statement about the number of H-bonds and have reworded our binding commentary to 'However, visualisation of these two predicted conformations (Supplementary Figure 12C and D) and a 2D interaction map (Supplementary Figure 12E and F) did not highlight any major differences in the types of interactions formed within the binding site. As such, although this covalent molecular docking suggests a possible binding mode, structural determination would be required to confirm compound binding mode and explain the differences in potency observed between the two enantiomers.' on p.17, lines 26-32.

Points from Reviewer 3:

Vuorinen et al. report a high throughput chemoproteomics platform capable of screening for DUB-targeted chemical probes. The authors demonstrate the efficacy of cysteine-directed electrophilic fragments as selective and potent starting points for chemical probes, despite their relatively simple chemical matter compared to larger drug-like compounds. In addition to increasing the screening throughput relative to previously reported studies, the authors utilized a high throughput medicinal chemistry-based library expansion approach to develop structure-activity relationships around hits from their primary screen. As proof of concept, the authors report a series of compounds from the primary and expansion screens with remarkable selectivity and potency for the OTU DUB subfamily, including the first reported inhibitor of OTUD7B. In summary, this paper successfully illustrates the utility of a high-throughput chemoproteomic screen coupled with a top-down, target-based validation and optimization scheme. Despite the growing interest in studying DUB biology, there is a lack of reported potent and selective DUB inhibitors across all subfamilies. This study builds on the growing body of DUB literature by reporting starting points for future inhibitors while also improving on previously reported platforms by adding a unique validation and optimization arm to their workflow. Additionally, by starting with a chemoproteomic screen, the authors alleviate concerns of promiscuity against DUBs for lead compounds which are characteristic of other approaches. While other papers have reported similar primary screening and validation approaches, this study is unique in that it started with commercially available electrophilic fragments which were rapidly elaborated following hit identification. Therefore, I recommend publication in Communications Chemistry with the following minor revisions.

We thank reviewer 3 for their supportive and positive comments about our work.

Specific Comments:

1) The current justification for screening only chloroacetamide warheads is insufficient. While it is accurate that acrylamides are less reactive than chloroacetamides, this is true for all target classes, not just DUBs. The authors would benefit from a more nuanced discussion about warhead selection. For instance, did the authors start with a more reactive warhead to overcome the lower affinity of fragment compounds? Would the authors plan to install a new warhead with better pharmacological properties if they found a preferential chemotype for a DUB of interest?

We thank reviewer 3 for highlighting this point. We have incorporated a more nuanced discussion on chloroacetamide electrophile selection in the main text on p.4+5, lines 32-34 and 1-4, 'Although the more intrinsically reactive chloroacetamide electrophile may not be optimal for clinical translation, there have been several examples in literature demonstrating that selectivity can be achieved with the chloroacetamide electrophile. Moreover, our aim was to develop potent tool compounds using a fragment-based approach, and employing the chloroacetamide electrophile provides a useful avenue into hit identification where fragments have modest reversible interactions.'

2) Along the same lines, the authors note that their lead compounds have high selectivity against the DUBome but there is no data against other target classes, which is a concern given the reactivity of chloroacetamides. fail to profile their compounds' reactivity against other proteins. The authors should consider a cysteine profiling experiment to highlight potential off targets, or at least acknowledge the importance of broader profiling before qualifying compounds as hits and probes down the line.

Thank you for raising this point, which was also highlighted by reviewer 2. As discussed in our response to reviewer 2, to address the proteome-wide selectivity of compounds 29 and 30, we have included an additional chemoproteomics cysteine profiling experiment, which uses an iodoacetamide-desthiobiotin (IA-DTB) competitive probe workflow, which we have recently published (biorxiv, <https://doi.org/10.1101/2024.07.25.605137>). A heatmap of compound off-targets is shown in Supplementary Figure 10D, and we have discussed non-DUB off-targets identified in this IA-DTB experiment in the main text on p.14, lines 1-14. We have also included a description of this workflow in the methods section on p.21-23, starting on p.21, line 28. In line with the precedent set in the above recent publication, we have also renamed all chemoproteomics IC50 values to TE50 values (where TE = target engagement).

3) It is insinuated the hit compounds target the catalytic cysteine of DUBs but this needs to be demonstrated using mass spectrometry.

Thank you to reviewer 3 for raising this point, which was also raised by reviewer 2. We agree, and as discussed in our response to reviewer 2, we performed an additional site identification experiment using fragment labelled purified DUB. After trypsin digestion the peptides were analysed by LC/MS-MS (added into Figure 5D), which confirmed the site of modification by compound 29 is the catalytic cysteine residue, Cys194. Unfortunately, we could not obtain this information for compound 30, due to the low labelling of compound 30 observed (as exemplified in additional kinetic characterisation experiment shown in Supplementary Figure 11B).

4) For the DUB biochemical assays, how were enzyme and substrate concentrations chosen? Was a titration conducted to ensure signal was in the linear range? Is the activity of the various enzymes similar? How long was compound preincubation time?

Thank you for this useful comment. We have added a more detailed description of the biochemical assay experimental set-up in the methods section on p.25, lines 19-25 and 27-30, 'Prior to inhibition assays, optimal conditions for each DUB were found by performing plate-based fluorescence time courses with a matrix of DUB and substrate dilutions. All assays were performed in 50 mM HEPES pH 7.5, 100 mM NaCl, 1 mM EDTA, 0.05% Tween20, 10 mM DTT. The reactions were initiated with the addition of Ub-Rho110Gly and monitored by measuring the rhodamine fluorescence every 30 s

at room temperature in a Clariostar Plus plate reader. Conditions were chosen so that the rate of reaction was within the linear phase.’ and ‘DUBs (2.5 μ M) were pre-treated with chosen fragments (0 – 200 μ M) for 3 h at room temperature. Following dilution of the DUBs, reactions were initiated with the addition of Ub-Rho110Gly and monitored as above.’

5) The authors should add a supplementary table detailing the coverage of each DUB reported in this study. It is essential to know how many unique peptides and PSMs are detected for each DUB in every sample to determine the reliability of the protein quantification.

Thank you for raising this helpful point. We have included this data in additional Supplementary Data files (SD1, SD2, SD3, and SD4) for each chemoproteomics experiment. This data can also be found in PRIDE, under identifiers PXD054883 and PXD057851 with log in details:

Username: reviewer_pxd054883@ebi.ac.uk

Password: Vfc1RvDICJ5z

And **Username:** reviewer_pxd057851@ebi.ac.uk

Password: zEIhur2C53fH

6) A supplementary table showing which of the three hit criteria each compound passed is necessary. Do compounds only need to pass 2/3 to be considered a hit? Currently, it is unclear if certain attributes are more heavily weighted than others, especially given the note that the log₂FC can be lower if the compound meets other criteria. If compounds must pass all three criteria, then there is no need for two log₂FC cutoffs.

Thank you for raising this point. We have included this supplementary table in Supplementary Figure 4C.

7) Supplementary figure 2 is incomplete or improperly described. Each of the three subfigures has a different number of fragments along the x-axis.

Thank you for highlighting this point, which was also highlighted by reviewer 1. As detailed in our response to reviewer 1, in Supplementary Figure 2 (now called Supplementary Figure 3), we have only shown the fragment hits which compete at least one DUB in that sub-family – we do not show all of the fragments in the library for ease of visualisation. Thus, there are different numbers of fragment hits identified within each sub-family represented in A) UCH DUB sub-family, B) USP DUB sub-family and C) MJD DUB sub-family. We have made this clearer in the figure caption and with the axis title in the figures. We have also numbered the fragment hits on the x-axis for clarity too. We have additionally included a new heatmap in Supplementary Figure 2, which shows the entire 227 fragment library against the 43 identified DUBs.

8) Regarding figure 2C, the authors should change the scale of the right y-axis to correspond to the left y-axis, perhaps by displaying engagement by percent compound labeling instead of Log₂FC. As currently depicted, 50% engagement ABPP corresponds to 25% labeling by intact protein MS.

Thank you for raising this point. We have changed the right-side y axis on Figure 2C to represent chemoproteomics percentage labelling as ‘100-Biotin-Ahx-Ub-VS probe

labelling', which we consider improves the visualisation and comparison of chemoproteomics and intact protein LC-MS data.

9) Spectral data were not available in PRIDE during the review of the manuscript. The authors should include spreadsheets containing the processed datasets for both ABPP and intact protein-MS experiments as supplemental files for ease of utilizing the data in this study.

Thank you for highlighting this. As mentioned above, we have included the ABPP data in additional Supplementary Data files (SD1, SD2, SD3, and SD4) for each chemoproteomics experiment. This data can also be found in PRIDE, under identifiers PXD054883 and PXD057851 with log in details:

Username: reviewer_pxd054883@ebi.ac.uk

Password: Vfc1RvDICJ5z

And **Username:** reviewer_pxd057851@ebi.ac.uk

Password: zEIhur2C53fH

The file size, and number of files are for the processed datasets for the intact protein-MS experiments, is incredibly large, and falls outside of the scope of the PRIDE database. However, we have added some representative examples of the deconvoluted intact protein LC-MS spectra for our *in vitro* validation (Supplementary Figure 6) and for our HTC-D2B screening (Supplementary Figure 7C) for each DUB.

10) Please report the starting amount of cell lysate used in the chemoproteomics experiments.

Thank you for highlighting this oversight – our apologies for not including this initially. We have added the starting amount of cell lysate used in chemoproteomics experiments (495 µg per sample) into the methods sections on p.20, line 21.

11) The authors state that a spectral library was generated using DIA data. This is odd, as spectral libraries are typically generated using DDA data acquired from the same or similar samples. If this is the case, the parameters used to acquire DDA data should be reported.

Thank you for raising this point. We did indeed generate the project specific spectral library using DIA data, which is supported by Pulsar search engine and Spectronaut. We have highlighted this in the methods section on p.21, line 10. There are no further parameters to add, as DDA data was not acquired.

We wish to thank the reviewers for their supportive comments and helpful suggestions for improving our manuscript. We agree with the majority of the reviewers' comments, and we have performed several new experiments and adapted our manuscript to address their points. We believe the revised manuscript is much improved as a result of their suggestions.

For clarity, we have addressed and responded to each point individually, in purple, below.

Points from Reviewer 1:

The manuscript by Vuorinen, Kennedy, McPhie et al. from the Bush and Rittinger groups describe an improved version of their previously reported chemoproteomics platform and the identification of covalent fragments binding to OTU deubiquitinases. The identification of novel and specific chemical matter for their targeting is in high demand due to the emerging therapeutic prospects of DUB inhibitors as well as DUB-recruiting bifunctional small molecules. The described platform uses different high-end mass spectrometry approaches (both for chemoproteomics as well as intact protein analysis, with notable improvements over the previous version, trading some coverage for speed) combined with high throughput chemistry, and thereby pushes the boundaries of the field. These approaches are well described and well-illustrated in the manuscript. The identification and initial characterization of an enantioselective fragment for OTUD7B validates the approach.

We thank reviewer 1 for their encouraging comments.

However, the manuscript suffers from several issues which preclude publication in its current form. These include missing chemical synthesis/characterization data, poor reference to the literature, technical inconsistencies rendering some data uninterpretable, some overstatements, and missing validation of the binding mode. In addition, the high degree of automated/processed data analysis makes it difficult to assess raw data quality. These points are described here in detail and should be addressed before publication can be considered:

We acknowledge the reviewer's comments and have addressed the detailed comments below.

- The authors mention in the text "Compounds 11 – 28 were successfully resynthesized, purified and taken forward". However, the manuscript lacks any descriptions of syntheses or chemical characterization data (1H, 13C, MS). In line with journal standards, these data must be added, including ee% for compounds 29/30 or a reference to the ee of their starting materials. Fig. S4 states that some compounds could not be made. If so, please rephrase the text accordingly.

We thank reviewer 1 for highlighting that our wording was unclear. All compounds used were purchased from Enamine (catalogue numbers are in Supplementary Table 1). Upon arrival, compounds were quality checked by LC-MS, and these QC purity values have been added into Supplementary Table 1. We have altered the text on p.11, line 5 in the main text to 'purchased as purified compounds', and p.24, lines 23-27 in the

methods section to 'Compounds 11 – 30 were purchased from Enamine. Internal compound quality control upon arrival of commercial order was carried out by LC-MS, and compound purity by LC-MS are reported in Supplementary Table 1.' to make this clear.

- Regarding the high throughput chemistry, the authors mention that understandably „product purities were not uniform across the plate“. It would be helpful if the authors could include a few exemplary LC-MS traces to demonstrate the properties of the material that was screened. These will not be completely pure compounds, but this information might be useful to others adopting the approach.

We have added Supplementary Figures 7A and 7B to demonstrate the distribution of conversion rates with HTC, and examples of LC-MS traces for each category of conversion (ie. 0%, <25%, 25-50%, >50% conversion).

- Also regarding the HTC-D2B assay, the text states “Following analysis by LC-MS to determine conversion rates⁴¹, and a reagent quench with hydroxylamine“ but the methods state that there was no quench performed for OTUD7B and ZRANB1. Why not? Fig. 3B shows these two DUBs have massively elevated hit rates, which the authors relate to their fitting selection of amines in the second library, when in fact (if there was no quench for some DUBs) this may be a technical artefact due to retained chemical reactivity. This may be particularly relevant for OTUD7B which has a very high propensity to react with electrophiles.

Thank you to reviewer 1 for raising this important point. We did not use a hydroxylamine quench for OTUD7B or ZRANB1 because both proteins were unstable to the hydroxylamine treatment, as observed by significant differences in the intact protein LC-MS traces. In contrast, the other DUBs tolerated the quench conditions well. It is possible that this affected the number of hits for OTUD7B and ZRANB1. Because of this, we were careful not to make comparisons of hit compounds from the HTC plates across the DUBs, only to contextualise hits for a given DUB. To clarify this, we have added the words 'where tolerated' on p.10, line 20, and the sentence 'OTUD4, OTUD5 and OTUD7A tolerated exposure to hydroxylamine well, however we observed a loss of MS signal for OTUD7B and ZRANB1, and screening of the HTC library was repeated against these two proteins without the reagent quench step' on p.10, lines 24-26. Additionally, we have added the sentence 'To avoid comparisons between labelling events with and without the hydroxylamine quench step, we focused on compounds that showed improved labelling for a given protein at this stage, rather than comparing fragment labelling across all five proteins.' on p.10, lines 31-34.

- The degree of labeling in this intact protein mass assay is used also to compare across different DUBs, but rarely 100% labeling is reached even at the high 200 μ M concentrations. Likewise, OTUD5 covalent labeling seems to be leveling off at around 60% (OTUD7B at around 90%), so there be some heterogeneity in protein batches which should be considered (maybe with a Ub-VS measurement, and normalization to the highest labeling observed there as the portion of catalytically competent enzyme). Otherwise comparisons between DUBs and regarding specificity have to be treated with caution.

Thank you for raising this point. There are unavoidable variabilities in heterogeneity across different recombinant DUB preps. As with the point above, we have avoided making comparisons of hits across different DUB labelling with our recombinant proteins, and have instead focused on comparison of hits for one given DUB. To clarify

this, we have added the sentence 'We observed that different DUBs reached differential maximum labelling percentages (Supplementary Figure 5A and B), suggesting heterogeneity between protein batches, and therefore direct comparisons between labelling percentages across DUBs were treated with caution' on p.8, lines 20-23.

- The high degree of processed data makes it difficult to assess the mass spec data quality. Please include a few deconvoluted mass spectra of different DUBs and fragments to illustrate the results.

Thank you for raising this point. We have added examples of deconvoluted spectra for our in vitro validation (Supplementary Figure 6) and for our HTC-D2B screening (Supplementary Figure 7C) for each DUB.

- The assay seems to have been carried out at 4°C – how does the reactivity change when the incubation is done at more relevant temperatures (e.g. RT or 37°C). This could be combined in a new figure, as this will demonstrate the utility of the fragments and may remove a bias towards the DUB with the highest remaining reactivity at 4°C. We have performed labelling experiments with recombinant proteins at 4°C in order to maximise protein stability over the 24-hour incubation period. We believe that screening at higher temperatures over a 24-hour incubation period would risk partial denaturation/precipitation of protein and hence affect the reproducibility of our experiments.

- The same point regarding the processed data can be made for the chemoproteomics. This reviewer understands the needs to compress the data into heatmaps, but 1-2 volcano plots should be included to (e.g. for the most advanced compounds) to allow readers to assess the data, also taking statistical significance of the results into account. Please include statements on the number of replicates for all mass spec experiments.

Thank you for this comment. We have added volcano plots for our chemoproteomics experiments with compounds 1 – 7 (Supplementary Figure 4) and compounds 29 and 30 (Supplementary Figure 10C) to address this point. We have also added the number of replicates (n=3 for UbVS chemoproteomics, and n=4 for IA-DTB chemoproteomics) to all relevant figure captions and in the Methods section.

- In Fig. 2A the scale starts at 1 (should actually be minus 1?), meaning that a fragment can inhibit a DUB up to 50% and the field is still white. This offset may make the data look cleaner, but gives a false impression of selectivity. The same is used for statements like „Did not label OTUD7A“ in relation to Fig. 2C, when several compounds label up to 40%.

This is a very good point. We have amended the scales for all heatmaps accordingly to start at 0 (Figure 2A, and Supplementary Figures 2 and 3), and have highlighted our use of absolute average log₂ values for clarity of visualisation in the figure captions. We have also changed the RHS y-axis on Figure 2C to represent chemoproteomics percentage labelling as '100-Biotin-Ahx-Ub-VS probe labelling', which we think improves the visualisation of data. We have also amended the statement 'did not label 7A' to 'For OTUD7A, chemoproteomics and intact protein LC-MS were less concordant, with some discrepancies between the two techniques observed for several fragments labelling OTUD7A' on p.8, lines 17-20.

- The data in Supp Fig. 3. e.g. on fragment 3 look very strong, and overall these are great data to build from. However, the authors should mention explicitly that fragment 2 was used in their previous publication (Cookson et al.), was termed fragment 26 there and was already there found to be highly OTUD7B reactivity (Fig. 5A+B).

Thank you for highlighting this. We have added the sentence 'The selected fragments included compound **2**, which was profiled as an OTUD7B inhibitor in our previous work³⁷ (therein compound **26**).' to p.8, lines 10-12, to draw attention to our previous work in more detail.

- OTUD7B is also named Cezanne 1, and Mission Therapeutics filed patents for highly potent, covalent Cezanne 1 inhibitors in 2015-16 (e.g. EP4067355A1, EP3394049B1). The latter includes an enantioselective pair of compounds 12: IC₅₀ < 0.1 uM; 13: IC₅₀ 1-20 uM, which mirrors the about 20x difference the authors observe for their compound. The compounds also have the same difference between stereocenter and electrophilic carbon. This should be included, and any statements like "first published chemical structure of an OTUD7B inhibitor" be adjusted.

Thank you for drawing attention to this oversight on our part. We have removed this claim from our manuscript and added the following sentences: to the introduction on p.2, lines 29-30 'A series of cyanopyrrolidine inhibitors of OTUD7B have also been disclosed in patents from Mission Therapeutics...'; and to the results and discussion on p.14, lines 20-22 'Interestingly, this observation concurs with enantioselectivity observed for a pair of published cyanopyrrolidine OTUD7B inhibitors¹⁹.'

- Likewise, the authors should reference recent and relevant work from the Geurink (HT chemistry for covalent probes, UCHL1 compounds in live cells), Gersch (chemogenomic probe pairs, specificity analysis), Flaherty (UCHL1 covalent fragment, fragment hit expansion) labs.

We have added references (numbers 13 – 15) to our manuscript, described in the sentence on p.2, lines 22-26 'In fact, UCHL1 has been the target of several successful covalent chemical probe discovery campaigns, where optimised cyanopyrrolidine and chloroacetamide activity-based probes have been used to characterise the mechanisms of deubiquitination and the downstream effects of UCHL1 inhibition in live cells and zebrafish embryos.'

- These references will also be helpful to put the obtained fragments into context. The authors use the word "probe" in the title (which should be avoided, as compound 29 is not nearly as specific or characterized to justify this designation). In the absence of a specificity or phenotype analysis in live cells (as done by all these groups for UCHL1 targeting probes / tool compounds), even "tool compound" / "biological tools" is a bit of stretch. These limitations should be addressed or openly discussed.

Thank you for this highlighting this point. We have changed the word 'probe' to 'fragment' in the title and abstract, and we have added the following sentence to the conclusion on p.18, lines 26-27 'a novel pharmacophore and the first use of a chloroacetamide electrophile for OTUD7B inhibition.'

- The curves for OTUD7A in Supp Fig. 6 look odd. As the specificity of the enantioselective fragment for OTUD7B is based on these data, can the authors comment on how confident they are on these data? The focus on abs. log₂ ratios for data visualization makes sense, but when it comes to specificity, this representation is

misleading (e.g. Fig. 4B: -2.5 means 83% labeled, -5.5 means 98% labeled; this looks like a big difference, but is not different enough for a cellular experiment).

With regards to the curves for OTUD7A in the now renumbered Supplementary Figure 10B, we agree that these are not ideal IC50 curves. However, the chemoproteomics data for OTUD7B gave robust concentration-response curves and small error bars per point, so we are confident in the OTUD7B data. We have added a sentence into the main text to address the OTUD7A chemoproteomics on p.12, lines 26-29 'Due to identification of a small number of low intensity peptides for OTUD7A, we were unable to measure accurate IC50 values for OTUD7A by chemoproteomics, and therefore could not compare compound selectivity between OTUD7B and OTUD7A by chemoproteomics.' We have instead addressed the question of selectivity over OTUD7A using biochemical assays (Figure 5A and Supplementary Figure 11A) and in vitro kinetics characterisation (Supplementary Figure 11B – this is an additional experiment performed to address reviewer 1 minor point, and reviewer 2 and 3 comments, see below for details) with enantiomeric compounds 29 and 30, which confirmed the specificity of these compounds for OTUD7B over OTUD7A.

With regards to the use of abs log2 ratios in Figure 4B (the 200 and 50 uM), this is a typical and broadly adopted method for displaying of proteomics data, particularly when analysing at a single concentration. However, we have also added in another representation of this data in the form of a volcano plot (Supplementary Figure 10C), which also allows visualisation of specificity in a different way. We agree with the reviewer that log2 scale is a crude method for performing a detailed comparison of potencies of compounds. It is for this reason that we ran selected compounds in full concentration response, and this data we have indeed represented as percentage labelling, in agreement with the reviewers suggestion, in Figure 4D and E (and Supplementary Figure 10B).

- The docking figures at the end capturing the bound state lack validation. The authors should attempt to validate the obtained poses by mutation. Firstly, by assessing that actually the catalytic cysteine is bound, and then secondly, e.g. by mutating Asp191 or Leu294 to assess whether these residues contribute to enantioselectivity as would be predicted by the shown geometries. A chemical validation (fragments with substituents at the phenyl ring should increase discrimination between stereoisomers) will likely be out of scope, but at the moment this is seems very preliminary (not even considering that docking the "post-reaction" state may not always capture what drives reaction speed). The catalytic cysteine may seem obvious but other fragments/compounds bind non-catalytic Cys in e.g. OTUB1 or UCHL1. Providing these data is important to justify the claim that the compound provides an "excellent starting point for further structure-based optimization".

Thank you for this comment. To validate the site of binding, ie the residue at which the chloroacetamide binds, we performed an additional site identification experiment using fragment labelled purified DUB. After trypsin digestion the peptides were analysed by LC/MS-MS (added into Figure 5D), which confirmed the site of modification by compound 29 is the catalytic cysteine residue, Cys194. This validates the residue used for covalent docking, however we have also put less emphasis on these docking experiments by moving the figures into Supplementary Figure 12 and adding the sentence on p.17, lines 29-32 'As such, although this covalent molecular docking suggests a possible binding mode, structural biology would be required to further elucidate compound mechanism of action and explain the differences in potency observed between the two enantiomers.' We have also removed the words 'structure-

based' from the sentence 'provides an excellent starting point for further structure-based optimization' on p.17, line 37. We believe that extensive mutagenesis and further SAR exploration is out of scope.

Minor points:

Thank you for very much for highlighting these minor points. We have addressed these points in the manuscript as outlined below.

- Supp Fig. 3C: The previous finding that OTUD5 phosphorylation mediates ubiquitin recognition (i.e. not nucleophilicity) could be added as this is in line with these results. We have added the sentences 'OTUD5 is known to be activated in cells through phosphorylation, altering its propensity to bind ubiquitin. To investigate whether we had selective fragment binders for either form of OTUD5, we validated both the inactive and activated forms, however we observed similar labelling for both states (Supplementary Figure 5C), supporting that OTUD5 phosphorylation does not alter the nucleophilicity of the catalytic cysteine.' on p.8, lines 28-33.

- The rationale for therapeutic targeting of OTU DUBs is oddly derived from their ability to cleave specific polyUb linkages without any further information, this should be rephrased or made stronger by direct disease relation. (Refs 17 and 18 include an X). The X in references 17 and 18 (now references 22 and 23) has been removed. We have also rephrased the paragraph you describe on p.3, lines 9-15, to improve the rationale for therapeutic targeting.

- „Only a handful of OTU DUBs have high-quality chemical tools“ requires a citation. I would also doubt that this statement is accurate.

We have rephrased this statement to say 'many members of the OTU DUBs have yet to be targeted by well-characterised chemical tools' on p.6, lines 2-3.

- „fragment hits against this family were in an optimal reactivity space“ What does this mean?

We have rephrased this statement to say 'We observed that some of these fragment hits labelled one OTU DUB selectively, while other fragments labelled several OTU DUBs with minimal labelling of other DUB families. Crucially, none of the fragment hits against OTU DUBs labelled more than 15 other non-OTU DUB family proteins, which suggested good protein ligandability without extensive fragment promiscuity (Supplementary Figure 3A, B and C).' on p.6, lines 6-10.

- In Supp Fig. 2, why are there fewer fragments (fewer columns) for A (ATXN3) and C (UBP DUBs) than in B (UCHs)?

In Supplementary Figure 2 (now called Supplementary Figure 3), we have only shown the fragment hits which compete at least one DUB in that sub-family – we do not show all of the fragments in the library for ease of visualisation. Thus, there are different numbers of fragment hits identified within each sub-family represented in A) UCH DUB sub-family, B) USP DUB sub-family and C) MJD DUB sub-family. We have made this clearer in the figure caption and with the axis title in the figures. We have also numbered the fragment hits on the x-axis for clarity too. We have also additionally included a new heatmap in Supplementary Figure 2, which shows the entire 227 fragment library against the 43 identified DUBs.

- IC50 values of covalent compounds heavily depend on the experimental conditions, and while frequently used, should be treated with care. Kobs/I values should be recorded at least for fragment 29+30 to allow cross-comparison to other studies.

To complement the IC50 values, we have performed full kinetics characterisation for compounds 29 and 30 against recombinant OTUD7A and OTUD7B (Figure 5B and C, and Supplementary Figure 11B), and we have reported k_{inact}/K_i values for compounds 29 and 30 against OTUD7B (Figure 5C, and in the main text). We were not able to calculate k_{inact}/K_i for compounds 29 and 30 against OTUD7A as the labelling we observed was too low. We describe this kinetic characterisation in the main text on p.15, lines 5-21.

- Supp Fig. 6A shows an IC50 of 1.6 uM and seems to relate to data shown in Fig. 4A, however, the data curves there suggest the 50% transition to occur more at 3.5 uM (was the upper 100% boundary fixed during the curve fitting)? This would also be consistent with the in vitro IC50, and the 20x reactivity difference.

The upper boundary was unconstrained in the original version of our manuscript, we have therefore recalculated all IC50 values and TE50 values with the upper boundary now set to top=100 in GraphPad Prism. We have changed the figures and highlighted this in the figure captions (Figure 4D and E, Figure 5A and Supplementary Figures 10B and 11A). We have adjusted the IC50 values in the figures and in the text accordingly. We have also highlighted the curve-fitting method in the methods section on p.21, lines 21-26 to include this change 'Concentration-response curves were fitted using label free protein quantity values. Values were normalised against samples treated with DMSO and plotted as percentages of Biotin-Ahx-Ub-VS labelling against fragment concentration using GraphPad Prism (v. 10). The curves were fitted using four parameter nonlinear regression with constraints bottom=0, top = 100. Each experiment was set up as technical triplicates.' For completeness, we have also reported the 95% confidence interval values as calculated using GraphPad Prism (v. 10), reported in Supplementary Figure 10A and 11A.

Points from Reviewer 2:

This manuscript by Vuorinen and coauthors described a chemoproteomics screening of a small library of chloroacetamide fragments targeting DUBs and the subsequent optimization of the hits through high-throughput chemistry and cell lysate-based screening using improved LC-MS workflow. This work demonstrated improved screening capacity of the cell lysate-based chemoproteomics workflow, which is a major strength of the manuscript. Further, an interesting enantioselective compound was identified with demonstrated selectivity toward OTUD7B. Although compound 29 is not yet ready to be used as a tool compound for OTUD7B, it provides a promising starting point for future scaffold expansion and optimization. Overall this is a well-written manuscript and will be of high interest to the DUB field. Below are points that should be addressed before the publication of this work.

We thank reviewer 2 for their enthusiasm for our work. We have addressed the points raised below.

The authors stated that “all fragments which targeted OTUD7A by chemoproteomics did not label OTUD7A by intact protein LC-MS”. It does not seem to be an accurate description of the results shown in Figure 2C regarding the OTUD7A in vitro and chemoproteomics data. For example, fragment 5 and 7 seem to label OTUD7A efficiently in both experiments, while fragment 1 showed good in vitro labeling but poor chemoproteomics labeling. Please clarify.

We thank this reviewer for raising this important point, which was also raised by reviewer 1. As described in the response to reviewer 1, we have changed the right-side y axis on Figure 2C to represent chemoproteomics percentage labelling as ‘100-Biotin-Ahx-Ub-VS probe labelling’, which we think improves the visualisation and comparison of chemoproteomics and intact protein LC-MS data. We have also amended the statement ‘did not label OTUD7A’ to ‘For OTUD7A, chemoproteomics and intact protein LC-MS were less concordant, with some discrepancies between the two techniques observed for several fragments labelling OTUD7A’ on p.8, lines 17-20.

The authors should include an SI figure similar to that in Figure 2A for other families of DUBs. This information would be useful to the readers and give this work broader significance.

Thank you for this helpful suggestion. In Supplementary Figure 3, we show heatmaps similar to that in Figure 2A, which show the fragment hits which compete at least one DUB in the three other DUB sub-families in A) UCH DUB sub-family, B) USP DUB sub-family and C) MJD DUB sub-family. We have now made this clearer in the figure caption and with the axis title in the figures. We have also numbered the fragment hits on the x-axis for clarity too. We have additionally included a new heatmap in Supplementary Figure 2, which shows the entire 227 fragment library against the 43 identified DUBs.

The authors need to demonstrate that compounds 29 and 30 label the catalytic cysteine of OTUD7B using purified DUB and trypsin-digestion MS analysis.

Thank you for raising this point. We performed an additional site identification experiment using fragment labelled purified DUB. After trypsin digestion the peptides were analysed by LC/MS-MS (added into Figure 5D), which confirmed the site of modification by compound 29 is the catalytic cysteine residue, Cys194. Unfortunately, we could not obtain this information for compound 30, due to the low labelling of compound 30 observed (as exemplified in additional kinetic characterisation experiment shown in Supplementary Figure 11B).

It is not entirely clear how the percentage quantification (such as those in Figure 4D & E) of Biotin-Ahx-Ub-VS labeling of DUBs in the cell lysates was done. Were LFQ values used to calculate the percentage of DUB being captured and detected by mass spectrometry? Please discuss this in the Methods section.

Thank you for highlighting this oversight. We have added a more detailed description of this into the methods section on p.21, lines 21-26.

One caveat of the above approach is that the Ub-VS probe may label different DUBs in the lysate with different affinity/potency. Thus the apparent IC50 obtained by fitting the curves in Figure 4D & E may not reflect accurately the potency of the compound. A more rigorous kinact/KI determination would be beneficial to confirm the selectivity of the compounds versus selected DUBs.

Thank you for raising this important point, which was also raised by reviewer 1. As described in the response to reviewer 1's comment, to complement the IC₅₀ values, we have performed full kinetics characterisation for compounds 29 and 30 against recombinant OTUD7A and OTUD7B (Figure 5B and C, and Supplementary Figure 11B), and we have reported k_{inact}/K_i values for compounds 29 and 30 against OTUD7B (Figure 5C, and in the main text). We were not able to calculate k_{inact}/K_i for compounds 29 and 30 against OTUD7A as the labelling we observed was too low. We describe this kinetic characterisation in the main text on p.15, lines 5-21.

The authors did not discuss the off-target labeling of proteins other than DUBs by the chloroacetamide compounds. This is an important consideration in developing covalent inhibitors targeting cysteine.

Thank you for highlighting this point. To address the proteome-wide selectivity of compounds 29 and 30, we have included an additional chemoproteomics experiment, which uses an iodoacetamide-desthiobiotin (IA-DTB) competitive probe workflow, which we have recently published (biorxiv, <https://doi.org/10.1101/2024.07.25.605137>). A heatmap of compound off-targets is shown in Supplementary Figure 10D, and we have discussed non-DUB off-targets identified in this IA-DTB experiment in the main text on p.14, lines 1-14. We have also included a description of this workflow in the methods section on p.21-23, starting on p.21, line 28. In line with the precedent set in the above recent publication, we have also renamed all chemoproteomics IC₅₀ values to TE₅₀ values (where TE = target engagement).

In the HTC steps, the amino coupling reaction was quenched with hydroxylamine for OTUD4, OTUD5 and OTUD7A but not for OTUD7B or ZRANB1. Why?

Thank you for raising this important point, which was also raised by reviewer 1. As discussed in the response to reviewer 1, we did not use a hydroxylamine quench for OTUD7B or ZRANB1 because both proteins were unstable to the hydroxylamine treatment, as observed by significant differences in the intact protein LC-MS traces. In contrast, the other DUBs tolerated the quench conditions well. It is possible that this affected the number of hits for OTUD7B and ZRANB1. Because of this, we were careful not to make comparisons of hit compounds from the HTC plates across the DUBs, only to contextualise hits for a given DUB. To clarify this, we have added the words 'where tolerated' on p.10, line 20, and the sentence 'OTUD4, OTUD5 and OTUD7A tolerated exposure to hydroxylamine well, however we observed degradation of OTUD7B and ZRANB1, and screening of the HTC library was repeated against these two proteins without the reagent quench step' on p.10, lines 24-26. Additionally, we have added the sentence 'To avoid comparisons between labelling events with and without the hydroxylamine quench step, we focused on compounds that showed improved labelling for a given protein at this stage, rather than comparing fragment labelling across all five proteins.' on p.10, lines 31-34.

The comparison of the docking of compound 29 and 30 to OTUD7B was simply based on the H-bonds. However, an equal number of H-bonds are found in both cases. It is thus unlikely the major factor of the selectivity. How different are the energetics of the two compounds binding to the OTUD7B active site? How about a steric clash? Figure 5B & C can include a space-filling model of the compounds.

We agree with the reviewer's points and have rephrased the discussion in the main text to put less emphasis on these docking experiments. We have moved the docking figures to Supplementary Figure 12 and have included a space-filling model in Supplementary Figure 12C and D. We have removed the statement about the number of H-bonds and have reworded our binding commentary to 'However, visualisation of these two predicted conformations (Supplementary Figure 12C and D) and a 2D interaction map (Supplementary Figure 12E and F) did not highlight any major differences in the types of interactions formed within the binding site. As such, although this covalent molecular docking suggests a possible binding mode, structural determination would be required to confirm compound binding mode and explain the differences in potency observed between the two enantiomers.' on p.17, lines 26-32.

Points from Reviewer 3:

Vuorinen et al. report a high throughput chemoproteomics platform capable of screening for DUB-targeted chemical probes. The authors demonstrate the efficacy of cysteine-directed electrophilic fragments as selective and potent starting points for chemical probes, despite their relatively simple chemical matter compared to larger drug-like compounds. In addition to increasing the screening throughput relative to previously reported studies, the authors utilized a high throughput medicinal chemistry-based library expansion approach to develop structure-activity relationships around hits from their primary screen. As proof of concept, the authors report a series of compounds from the primary and expansion screens with remarkable selectivity and potency for the OTU DUB subfamily, including the first reported inhibitor of OTUD7B. In summary, this paper successfully illustrates the utility of a high-throughput chemoproteomic screen coupled with a top-down, target-based validation and optimization scheme. Despite the growing interest in studying DUB biology, there is a lack of reported potent and selective DUB inhibitors across all subfamilies. This study builds on the growing body of DUB literature by reporting starting points for future inhibitors while also improving on previously reported platforms by adding a unique validation and optimization arm to their workflow. Additionally, by starting with a chemoproteomic screen, the authors alleviate concerns of promiscuity against DUBs for lead compounds which are characteristic of other approaches. While other papers have reported similar primary screening and validation approaches, this study is unique in that it started with commercially available electrophilic fragments which were rapidly elaborated following hit identification. Therefore, I recommend publication in Communications Chemistry with the following minor revisions.

We thank reviewer 3 for their supportive and positive comments about our work.

Specific Comments:

1) The current justification for screening only chloroacetamide warheads is insufficient. While it is accurate that acrylamides are less reactive than chloroacetamides, this is true for all target classes, not just DUBs. The authors would benefit from a more nuanced discussion about warhead selection. For instance, did the authors start with a more reactive warhead to overcome the lower affinity of fragment compounds? Would the authors plan to install a new warhead with better pharmacological properties if they found a preferential chemotype for a DUB of interest?

We thank reviewer 3 for highlighting this point. We have incorporated a more nuanced discussion on chloroacetamide electrophile selection in the main text on p.4+5, lines 32-34 and 1-4, 'Although the more intrinsically reactive chloroacetamide electrophile may not be optimal for clinical translation, there have been several examples in literature demonstrating that selectivity can be achieved with the chloroacetamide electrophile. Moreover, our aim was to develop potent tool compounds using a fragment-based approach, and employing the chloroacetamide electrophile provides a useful avenue into hit identification where fragments have modest reversible interactions.'

2) Along the same lines, the authors note that their lead compounds have high selectivity against the DUBome but there is no data against other target classes, which is a concern given the reactivity of chloroacetamides. fail to profile their compounds' reactivity against other proteins. The authors should consider a cysteine profiling experiment to highlight potential off targets, or at least acknowledge the importance of broader profiling before qualifying compounds as hits and probes down the line.

Thank you for raising this point, which was also highlighted by reviewer 2. As discussed in our response to reviewer 2, to address the proteome-wide selectivity of compounds 29 and 30, we have included an additional chemoproteomics cysteine profiling experiment, which uses an iodoacetamide-desthiobiotin (IA-DTB) competitive probe workflow, which we have recently published (biorxiv, <https://doi.org/10.1101/2024.07.25.605137>). A heatmap of compound off-targets is shown in Supplementary Figure 10D, and we have discussed non-DUB off-targets identified in this IA-DTB experiment in the main text on p.14, lines 1-14. We have also included a description of this workflow in the methods section on p.21-23, starting on p.21, line 28. In line with the precedent set in the above recent publication, we have also renamed all chemoproteomics IC50 values to TE50 values (where TE = target engagement).

3) It is insinuated the hit compounds target the catalytic cysteine of DUBs but this needs to be demonstrated using mass spectrometry.

Thank you to reviewer 3 for raising this point, which was also raised by reviewer 2. We agree, and as discussed in our response to reviewer 2, we performed an additional site identification experiment using fragment labelled purified DUB. After trypsin digestion the peptides were analysed by LC/MS-MS (added into Figure 5D), which confirmed the site of modification by compound 29 is the catalytic cysteine residue, Cys194. Unfortunately, we could not obtain this information for compound 30, due to the low labelling of compound 30 observed (as exemplified in additional kinetic characterisation experiment shown in Supplementary Figure 11B).

4) For the DUB biochemical assays, how were enzyme and substrate concentrations chosen? Was a titration conducted to ensure signal was in the linear range? Is the activity of the various enzymes similar? How long was compound preincubation time?

Thank you for this useful comment. We have added a more detailed description of the biochemical assay experimental set-up in the methods section on p.25, lines 19-25 and 27-30, 'Prior to inhibition assays, optimal conditions for each DUB were found by performing plate-based fluorescence time courses with a matrix of DUB and substrate dilutions. All assays were performed in 50 mM HEPES pH 7.5, 100 mM NaCl, 1 mM EDTA, 0.05% Tween20, 10 mM DTT. The reactions were initiated with the addition of Ub-Rho110Gly and monitored by measuring the rhodamine fluorescence every 30 s

at room temperature in a Clariostar Plus plate reader. Conditions were chosen so that the rate of reaction was within the linear phase.’ and ‘DUBs (2.5 μ M) were pre-treated with chosen fragments (0 – 200 μ M) for 3 h at room temperature. Following dilution of the DUBs, reactions were initiated with the addition of Ub-Rho110Gly and monitored as above.’

5) The authors should add a supplementary table detailing the coverage of each DUB reported in this study. It is essential to know how many unique peptides and PSMs are detected for each DUB in every sample to determine the reliability of the protein quantification.

Thank you for raising this helpful point. We have included this data in additional Supplementary Data files (SD1, SD2, SD3, and SD4) for each chemoproteomics experiment. This data can also be found in PRIDE, under identifiers PXD054883 and PXD057851 with log in details:

Username: reviewer_pxd054883@ebi.ac.uk

Password: Vfc1RvDICJ5z

And **Username:** reviewer_pxd057851@ebi.ac.uk

Password: zEIhur2C53fH

6) A supplementary table showing which of the three hit criteria each compound passed is necessary. Do compounds only need to pass 2/3 to be considered a hit? Currently, it is unclear if certain attributes are more heavily weighted than others, especially given the note that the log₂FC can be lower if the compound meets other criteria. If compounds must pass all three criteria, then there is no need for two log₂FC cutoffs.

Thank you for raising this point. We have included this supplementary table in Supplementary Figure 4C.

7) Supplementary figure 2 is incomplete or improperly described. Each of the three subfigures has a different number of fragments along the x-axis.

Thank you for highlighting this point, which was also highlighted by reviewer 1. As detailed in our response to reviewer 1, in Supplementary Figure 2 (now called Supplementary Figure 3), we have only shown the fragment hits which compete at least one DUB in that sub-family – we do not show all of the fragments in the library for ease of visualisation. Thus, there are different numbers of fragment hits identified within each sub-family represented in A) UCH DUB sub-family, B) USP DUB sub-family and C) MJD DUB sub-family. We have made this clearer in the figure caption and with the axis title in the figures. We have also numbered the fragment hits on the x-axis for clarity too. We have additionally included a new heatmap in Supplementary Figure 2, which shows the entire 227 fragment library against the 43 identified DUBs.

8) Regarding figure 2C, the authors should change the scale of the right y-axis to correspond to the left y-axis, perhaps by displaying engagement by percent compound labeling instead of Log₂FC. As currently depicted, 50% engagement ABPP corresponds to 25% labeling by intact protein MS.

Thank you for raising this point. We have changed the right-side y axis on Figure 2C to represent chemoproteomics percentage labelling as ‘100-Biotin-Ahx-Ub-VS probe

labelling', which we consider improves the visualisation and comparison of chemoproteomics and intact protein LC-MS data.

9) Spectral data were not available in PRIDE during the review of the manuscript. The authors should include spreadsheets containing the processed datasets for both ABPP and intact protein-MS experiments as supplemental files for ease of utilizing the data in this study.

Thank you for highlighting this. As mentioned above, we have included the ABPP data in additional Supplementary Data files (SD1, SD2, SD3, and SD4) for each chemoproteomics experiment. This data can also be found in PRIDE, under identifiers PXD054883 and PXD057851 with log in details:

Username: reviewer_pxd054883@ebi.ac.uk

Password: Vfc1RvDICJ5z

And **Username:** reviewer_pxd057851@ebi.ac.uk

Password: zEIhur2C53fH

The file size, and number of files are for the processed datasets for the intact protein-MS experiments, is incredibly large, and falls outside of the scope of the PRIDE database. However, we have added some representative examples of the deconvoluted intact protein LC-MS spectra for our *in vitro* validation (Supplementary Figure 6) and for our HTC-D2B screening (Supplementary Figure 7C) for each DUB.

10) Please report the starting amount of cell lysate used in the chemoproteomics experiments.

Thank you for highlighting this oversight – our apologies for not including this initially. We have added the starting amount of cell lysate used in chemoproteomics experiments (495 µg per sample) into the methods sections on p.20, line 21.

11) The authors state that a spectral library was generated using DIA data. This is odd, as spectral libraries are typically generated using DDA data acquired from the same or similar samples. If this is the case, the parameters used to acquire DDA data should be reported.

Thank you for raising this point. We did indeed generate the project specific spectral library using DIA data, which is supported by Pulsar search engine and Spectronaut. We have highlighted this in the methods section on p.21, line 10. There are no further parameters to add, as DDA data was not acquired.